# Mutational analysis of *Phanerochaete chrysosporium´*s purine transporter

**Mariana Barraco-Vega**[1]*, **Manuel Sanguinetti**[2]*, **Gabriela da Rosa**[3], **Gianna Cecchetto**[4]

**1** Microbiología, Departamento de Biociencias, Facultad de Química Universidad de la República, Montevideo, Uruguay, **2** Sección Bioquímica, Facultad de Ciencias, Universidad de la República, Montevideo, Uruguay, **3** Departamento de Ciencias Biológicas, CENUR-Litoral Norte, Universidad de la República, Montevideo, Uruguay, **4** Microbiología, Instituto de Química Biológica, Facultad de Ciencias— Facultad de Química, Universidad de la República, Montevideo, Uruguay

* mariveba@gmail.com (MBV); msanguinetti@fcien.edu.uy (MS)

## Abstract

We present here a mutational analysis of the purine transporter from *Phanerochaete chrysosporium* (PhZ), a member of the AzgA-like subfamily within the Nucleobase Ascorbate Transporters family. We identified key residues that determine its substrate specificity and transport efficiency. Thirteen PhZ mutants were generated and heterologously expressed in *Aspergillus nidulans*. The growth of mutant strains in the presence of purines and toxic analogues and the uptake rate of radiolabelled hypoxanthine were evaluated. Results revealed that ten mutants showed differences in transport compared to the wild-type PhZ: six mutants completely lost function, two exhibited decreased transport activity, and two showed increased hypoxanthine uptake. Subcellular localization and expression level analyses indicated that the differences in transport activity were not due to trafficking issues to the plasma membrane or protein stability. A three-dimensional model of PhZ, constructed with the artificial intelligence-based AlphaFold2 program, suggested that critical residues for transport are located in transmembrane segments and an internal helix. In the latter, the A418 residue was identified as playing a pivotal role in transport efficiency despite being far from the putative substrate binding site, as mutant A418V showed an increased initial uptake efficiency for the transporter´s physiological substrates. We also report that residue L124, which lies in the putative substrate binding site, plays a critical role in substrate transport, emerging as an additional determinant in the transport mechanism of this family of transporters. These findings underscore the importance of specific residues in AzgA-like transporters and enhance our understanding of the intricate mechanisms governing substrate specificity and transport efficiency within this family.

## Introduction

Purine and pyrimidine nucleobases, besides being the precursors of nucleic acid biosynthesis, are also involved in cell signalling, homeostasis, nutrition, response to stress, and last but not least, energy metabolism. The external supply of purines can be vital to cells that either rely on

**Data Availability Statement:** All relevant data are within the manuscript and its Supporting Information files.

**Funding:** Agencia Nacional de Investigación e Innovación under the codes

PR_FCE_3_2013_1_100659 (MBV) and POS_1097576 (MBV). Comisión Académica de Posgrado CAP_2019 (MBV) The funders had no role in study design, data collection and analysis, decision to publish, or preparation of the manuscript.

salvage pathways for nucleotide synthesis or use purines as nitrogen or carbon sources through catabolism. Entry of these solutes is regulated by a range of different membrane transport systems depending on the type of cell. Microbial cells in all domains of life possess specific nucleobase transporters, usually distinct from those involved in the uptake of nucleosides [1–4].

Fungi possess four specific nucleobase uptake systems that belong to two structurally and evolutionary distinct protein families [2, 5] (http://www.tcdb.org/). These are the NAT family (Nucleobase Ascorbate Transporters, also called Nucleobase Cation Symporters 2 or NCS2) and the NCS1 (nucleobase cation symporter 1) family. Both families include two sub-groups (sub-families), with totally non-overlapping specificities. The NAT family has the UapA/C-like and the AzgA-like subfamilies and NCS1 has the Fcy-like and the Fur-like transporters. All these transporters function as H$^+$-dependent secondary transporters or H$^+$-symporters. NAT transporters are present in all major taxa of life. Among these, the UapA/C subfamily has been identified even in mammals, while the AzgA-like subfamily lacks homologs in animals [6–8].

UapA/C-like transporters are specific for uric acid and xanthine, whereas AzgA-like transporters are specific for adenine-hypoxanthine-guanine. In addition, UapA/C-like and AzgA-like transport different purine analogues and drugs, namely oxypurinol and allopurinol versus 8-azaguanine and 6-mercaptopurine, respectively [9].

Structural approaches such as X-ray crystallography and relative modelling have confirmed the classification of nucleobase transporters into two major families, initially based on functional assays [10]. The sub-group classification within the NAT and NCS1 families is supported by notable differences in substrate specificity and the presence of characteristic amino acid sequence motifs. A seemingly paradoxical observation is that, despite their structural similarity (both belong to the NAT family), UapA/C-like and AzgA-like proteins transport entirely distinct substrates and drugs [9].

The UapA/C-like subfamily has been extensively characterized through the functional determination of numerous proteins from different domains and kingdoms, mainly the *Aspergillus nidulans* UapA transporter. This Ascomycota has two transporters of this type, UapA, and UapC, which give their name to the subfamily. UapA is one of the most studied eukaryotic transporters, with its crystallographic structure available, and a wide range of mutants have been analysed, allowing the identification of key residues for its functionality and specificity [10–13].

When it comes to structure-function relationships, AzgA-like proteins are much less characterized. Given the low primary sequence similarity that these proteins have with UapA/C-like proteins and the phylogenetic analyses that place them in separate clusters, the proteins initially considered to be part of the NAT family were only the UapA/C-like ones. Since the discovery of AzgA in *A. nidulans* (the first AzgA-like protein) and for over a decade, AzgA-like proteins were grouped as an independent family [14]. Subsequently, structural analyses (performed through homology modelling) of AzgA and AzgA-like proteins from *Escherichia coli* suggested that despite the low similarity in primary sequence and phylogenetic distance, AzgA-like transporters are structurally similar to NATs [7, 15]. This was recently confirmed by the resolution of the cryo-electron microscopy structure of *Arabidopsis thaliana* AzgA-like transporter, AZG1, which showed that it shares a similar topology and domain arrangement with NATs [16]. For this reason, they are currently grouped as two subfamilies within the NAT family [9]. The results obtained with AzgA as well as with AzgA-like proteins from *E. coli* indicate that although AzgA-like and UapA/C-like proteins seem to share a common molecular ancestor (reflected in similar general and local topology), they also appear to have diverged significantly in their specificities by employing different polar side chains for substrate binding and transport [7, 15]. Since the substrate specificity of a protein arises from its amino acid

sequence, to elucidate the key elements that determine why proteins with the same structure have different substrate specificities, it is necessary to focus on the differences they present in their primary sequence. In this sense, Papakostas and colleagues (2013) [15] performed multiple protein sequence alignments of functionally known homologs and identified 6 highly conserved motifs (located in transmembrane segments, TMS: 1, 3, 5, 8, 9, 10 and 12) and 45 invariant residues that are only present among AzgA-like proteins. To initiate a rationally designed mutagenesis study in this subfamily, they selected functionally important residues in well-studied UapA/C-like proteins that are located in characteristic sequence motifs of the AzgA-like family as targets for replacement. The authors concluded that despite the weak sequence similarity between these two groups of proteins, they use distinct but topologically equivalent side chains to dictate the binding site function and selectivity [15]. Krypotou and colleagues (2014) [7] arrived at the same conclusion after performing a mutational analysis on the *A. nidulans* AzgA transporter. This analysis, combined with the molecular modelling of AzgA based on the *E. coli* uracil transporter UraA (a member NAT family) [1], allowed the identification of several residues critical for purine binding and/or transport in TMS3, TMS8 and TMS10. In particular, N131 (TMS3), D339 (TMS8) and E394 (TMS10) were proposed to directly interact with substrates, while D342 (TMS8) was proposed to be involved in substrate translocation. The importance of these residues was reflected by the fact that even highly conserved substitutions of these residues lead to significantly modified affinity constants for substrates. Of note, these residues were identical, or very similar to those shown to be irreplaceable for purine uptake activity or substrate selectivity in *E. coli* AzgA-like transporters [7, 15].

Our group have previously characterized the AzgA-like purine transporter of the Basidiomycota *Phanerochaete chrysosporium*, PhZ, by its heterologous expression in *A. nidulans* [17]. Of note, this was the first study of an AzgA-like purine transporter in Basidiomycota. The PhZ sequence (578 amino acids) shows high similarity to *A. nidulans* AzgA (58% identity, 72% similarity and 98% coverage). PhZ transports the same substrates as AzgA (i.e., adenine, hypoxanthine, guanine and the toxic purine analogue 8-azaguanine) and undergoes the same post-translational regulation (responding to the endocytosis mechanism in response to a primary nitrogen source), but they differ in substrate transport and/or recognition processes, resulting in varying efficiencies of transport when expressed in the same genetic (and therefore metabolic) context [17]. This led us to hypothesize that the observed differences in transport could be the result of differences in key residues between these two transporters.

To test this hypothesis, we first expanded the characterisation of the distinctive motifs of the AzgA-like family by increasing the number of sequences analysed. Our analysis encompassed not only protein sequences of known function (11 AzgA-like proteins) but also 144 sequences of hypothetical proteins available in the databases to cover all taxonomic ranges where homologs of these transporters have been identified. We then focused on those residues that differed between AzgA and PhZ in these motifs and selected them for mutagenesis. The mutational analysis was performed on the heterologously expressed PhZ in *A. nidulans* [17]. This allowed us to work in a known context since *A. nidulans*, in addition to having characteristics that allow the development of numerous molecular tools for the study of transporters [18], is the organism where both the catabolism and the purine uptake system have been studied in greater detail [11, 14, 19, 20]. In this way, all mutant versions of PhZ could be compared not only with the wild-type version of PhZ but also with AzgA from *A. nidulans*. Since all proteins were expressed in the same genetic and therefore metabolic context, the differences in transport obtained can be attributed to the differences in key residues that one protein has compared to the other.

Of note, during this study, we obtained additional mutants through random mutagenesis, which were also analysed. The overall mutational analysis allowed us to provide evidence of the functional relevance of additional residues located in TMS3 (L124 and S133, especially the

former, as it localises in the putative substrate binding site) and TMS10 (T392). Most importantly, we obtained experimental evidence supporting a critical role for transport of an internal helix between TMS11-12 in an AzgA-like transporter.

## Materials and methods

### Strains, media, growth conditions and transformation procedures

*Aspergillus nidulans* strains used in this study are listed in the S1 Table. The media, growth conditions and genetic techniques for *A. nidulans* were according to Pontecorvo et al. (1953) and Scazzocchio et al. (1982) [21, 22]. Growth tests were carried out in Minimal Medium (MM) at 37˚C. Gene symbols are defined in Clutterbuck (1973) [23]. Transformations of *A. nidulans* were carried out following the method of Szewczyk et al. (2006) [24].

Media and supplemented auxotrophies were at the concentrations given in http://www.fgsc.net. Nitrogen sources were used at the final concentrations: 5 mM ammonium L(+) tartrate ($(NH_4)_2C_4H_4O_6$), 10 mM sodium nitrate ($NaNO_3$), 0.5 mM purines (adenine, guanine, hypoxanthine, xanthine and uric acid) (Sigma-Aldrich, Merck). Toxic analogues were used at the final concentration of 0.7 mM in the presence of a nitrogen source: 0.8 mM ammonium tartrate for 8-azaguanine and 10 mM sodium nitrate for oxypurinol.

### Generation of PhZ mutants

Changes in the coding sequence of *phZ* were introduced by site-directed mutagenesis using the QuickChange method [25], with the plasmid pMB02 serving as the template [17]. The resulting plasmids were sequenced, and once verified, they were used as a template for amplifying the entire cDNA. To express the mutant variants and the wild-type protein in the same genetic context, a cassette was constructed for each mutation by fusing the coding sequence of *phZ* with the 5' and 3' UTRs of *azgA*. The cassette construction also included fusion with the gene encoding GFP (*gfp*) and a gene complementing riboflavin auxotrophy for subsequent selection of transformants (*riboB^AF* of *Aspergillus fumigatus*)). Each cassette was constructed using 3 independent PCRs and a Fusion-PCR [24]. The fusion products (approximately 6750 bp) were purified with the QIAquick PCR Purification Kit (QIAGEN) and used to transform the *A. nidulans* strain MV060 (named ΔZAC, S1 Table) lacking the main purine transporters *ΔazgA*, *ΔuapA* and *ΔuapC*. We used PCR conditions described in [17], primers are listed in the S2 Table. The *A. nidulans* strains expressing PhZ with the changes: L124M, T131A, S133T, I388V, A391G, and T392A were generated using the earlier method. Additionally, mutations Y54G, V58A, A128F, Y129D, A148V, A418V, and T429P were analysed, which emerged randomly due to DNA polymerase errors during the construction of transformation cassettes. Initially, they were evaluated as double mutants alongside another selected change in a preliminary screening (phenotypic). Subsequently, based on the results obtained, single mutants were constructed following the same work scheme used for the rest of the selected residues.

The number of insertions for each construction was determined by relative qPCR using the gamma actin gene (*actA*) as internal control and 10 ng of genomic DNA as a template [17, 26]. We used PCR conditions and primers previously described in Barraco-Vega et al. [17]. A monocopy strain of each variant was selected for further work. The *phZ* gene of the selected transformants was sequenced to verify the presence of only the desired changes.

### Epifluorescence microscopy

Samples for fluorescence microscopy were prepared as previously described [27]. Briefly, the samples were incubated on coverslips in liquid medium supplemented with $NaNO_3$ as

nitrogen source and the necessary supplements regarding auxotrophies, for 14 h at 28°C. Coverslips with germinated conidia were observed on a Nikon Eclipse 80i epifluorescent microscope with appropriate filters. The resulting images were acquired with a DS-Fi1 (Nikon) digital camera using NIS-Elements F3.0 software. Images were then processed with Adobe Fireworks CS4 software.

### Radiolabelled hypoxanthine uptake measurements

Uptake assays of [2,8-$^3$H]-hypoxanthine (19.5 Ci/mmol; Moravek Biochemicals) were performed with spores of *A. nidulans* strains in germination following the protocol recommended by Krypotou and Diallinas [18]. 25 mL of Minimal Medium with $NaNO_3$ and corresponding nutritional supplements were inoculated with $10^8$ spores and incubated for 3.5 hours at 37°C, 140 rpm. To determine this time, the germination of the PhZwt strain was microscopically evaluated at 120, 180, 195, 210, and 240 minutes. 210 minutes was the time just before the appearance of the germ tube. Reactions were carried out with a concentration of 0.7 μM for [2,8-$^3$H]-hypoxanthine at 37°C, and each condition was assayed in triplicate. We used Ultima Gold XR scintillation fluid (Perkin Elmer) and Microbeta Trilux scintillation counter recording the average counts obtained in 1 minute. Previously, different reaction times (15 seconds, 30 seconds, 1 minute, 2 minutes, 4 minutes, 8 minutes and 16 minutes) were evaluated, determining that the uptake was linear at one minute. As recommended by Krypotou and Diallinas [18], $K_{m/i}$ was determined as the $K_i$ when the competing substrate was the same compound (unlabeled hypoxanthine). In this context, $K_{m/i}$ reflects the initial uptake rate and can be considered equivalent to $K_m$, representing the affinity of the transporter for the substrate. Uptake assays were performed using a mixture with a fixed concentration of labelled hypoxanthine (0.2 μM, optimally 10–50 times lower than $K_{m/i}$) and an excess amount of unlabelled substrate (1 mM), or in the case of $K_{m/i}$ determination varying concentrations of unlabelled hypoxanthine (0.1 μM—1 mM). The $K_{m/i}$ values were calculated from the IC50 value (the concentration required to achieve 50% inhibition) using GraphPad Prism software from dose-response curves. In all cases, the Hill coefficient was close to -1 (consistent with the presence of a binding site). $K_{m/i}$ was considered equivalent to IC50, based on the equation $K_i = IC50 / 1 + [S] / K_m$, where [S] is the fixed concentration of radiolabelled substrate used (at least 10 times lower than the $K_m$ value). The counts obtained were converted into moles of substrate/min/$10^8$ conidia, based on the concentration and specific activity (19.5 Ci/mmol) of the radiolabelled substrate used. Background uptake values were corrected by subtracting values measured in the ΔZAC strain. All transport assays were conducted in two independent experiments, with three replicates for each concentration or condition. Statistical analyses were performed using JASP (version 0.19). A one-way analysis of variance (ANOVA) was used to assess differences. In cases where the Shapiro-Wilk test returned p-values below 0.05, indicating non-normality, non-parametric tests were applied. The Kruskal-Wallis test was used for group comparisons when the normality assumption was violated. Post-hoc analyses were performed using Tukey's Honest Significant Difference (HSD) test for parametric data and Dunn's test with Bonferroni and Holmes correction for non-parametric data (p-value < 0.05). Graphs were generated within GraphPad Prism software.

### Protein extraction and western blot analysis

Total proteins of 200 mg of grinded mycelia were extracted as described by Apostolaki et al. [28] from cultures grown in MM supplemented with nitrate at 30°C for 14 hours. Protein concentration was determined by the Pierce BCA Protein Assay kit (Thermo Fisher). Total proteins (50 μg) were separated by SDS-PAGE (10% (w/v) polyacrylamide gel) and electroblotted

(OmniPAGE Electroblotting Units, Cleaver Scientific Ltd) onto a nitrocellulose membrane (0.45 μm, Thermo Scientific). The membrane was then incubated in stripping buffer (10 mM Tris–HCl pH 6.8, 2% SDS and 100 mM β-mercaptoethanol) for 15 min at 55˚C (as described by Kaur & Bachhawat [29]). The membrane was then treated with 5% non-fat dry milk, and immunodetection was done with a primary mouse anti-GFP monoclonal antibody (Roche Applied Science), or a mouse anti-tubulin monoclonal antibody (clone AA4.3, Developmental Studies Hybridoma Bank, Iowa, USA), and a secondary sheep anti-mouse IgG HRP-linked antibody (GE Healthcare). Blots were developed using the Amersham ECL Western blotting detection reagents and analysis system (GE Healthcare), and images were acquired using the GENESYS software from the GBoxChemi XT4 System (Syngene).

### Tridimensional models construction

The 3D structure of proteins was modelled with AlphaFold2 using the recommended default settings [30]. Once the prediction was completed, five models were generated, ranked by their pLDDT (Predicted Local Distance Difference Test) and pTM (Predicted Templated Modeling) scores. The model with the highest scores (pLDDT = 85.5, pTM = 0.833) was selected for further analysis. The stereochemical quality, including the assessment of Ramachandran plots and other structural parameters, was evaluated using PROCHECK from the SAVES server (UCLA, Los Angeles, CA).

With the best model structure of PhZ (wt and L124M), we performed molecular docking assays using the Autodock tools to evaluate its interactions with purines [31]. The structures of the purines used as ligands (hypoxanthine and 8-azaguanine) were obtained from the Protein Data Bank database. Once the protein and ligands were prepared, a cubic grid of 20 Å was centered on the visible cavity between the core domain and the gate. Docking was performed using the Lamarckian genetic algorithm. Among numerous generated conformations, the top ten were analysed and ranked based on interaction energy criteria, and visualized using the programs VMD, Pymol (Schrödinger, LLC) and LigPlot [32, 33]. The arrangement of the hypoxanthine and 8-azaguanine molecules represents 10 out of 10 conformations with interaction energy ranging from -6.95.5 to -5.30 kcal/mol.

## Results

### Rational design and construction of PhZ mutants

To further analyse the characteristic motifs of the AzgA-like family, we carried out a multiple protein sequence alignment with 155 sequences from all taxonomic ranges (S1 Appendix). This includes 11 AzgA-like proteins with known functions (Fig 1), as well as 144 sequences of hypothetical proteins deduced from sequenced genomes available in the National Center for Biotechnology Information database (29 from prokaryotes, 35 from plants, 54 from ascomycete fungi, and 26 from basidiomycete fungi). Thirty-five invariant residues and four highly conserved motifs were identified: i) E-X2-[AG]-[GA]-X-[ATV]-T-[FW]-X-[ATS]-M-X-Y-[IS]-[ILV]-X-V-N (Motif 1), ii) P-X-[AGS]-X-[AG]-[PSC]-[GA]-[MLI]-[GS]-X-[NT]-A-[YF]-X-[AT]-[YF]-X2-V (Motif 2), iii) G-I-G-X-[FY]-[LI]-X3-[GA] (Motif 3), and iv) [FY]-[IV]-E-S-X-[AST]-G-X2-[EAV]-G-G-[RKA]-T-G (Motif 4) (X represents any amino acid). The greatest diversity in these motifs is found in sequences of prokaryotic origin; if only fungal and plant sequences are considered, the motifs are more conserved (S1 Fig). According to the substrate binding models proposed for AzgA, motifs 1 and 3 are in TMS1 (core domain) and TMS5 (gate domain) respectively, while motif 2 is located in TMS3 facing the cell exterior, and motif 4 in TMS10 facing the cell interior [7]. Under the hypothesis that analysing these motifs may reveal key residues involved in the function of AzgA-like proteins, we chose to investigate

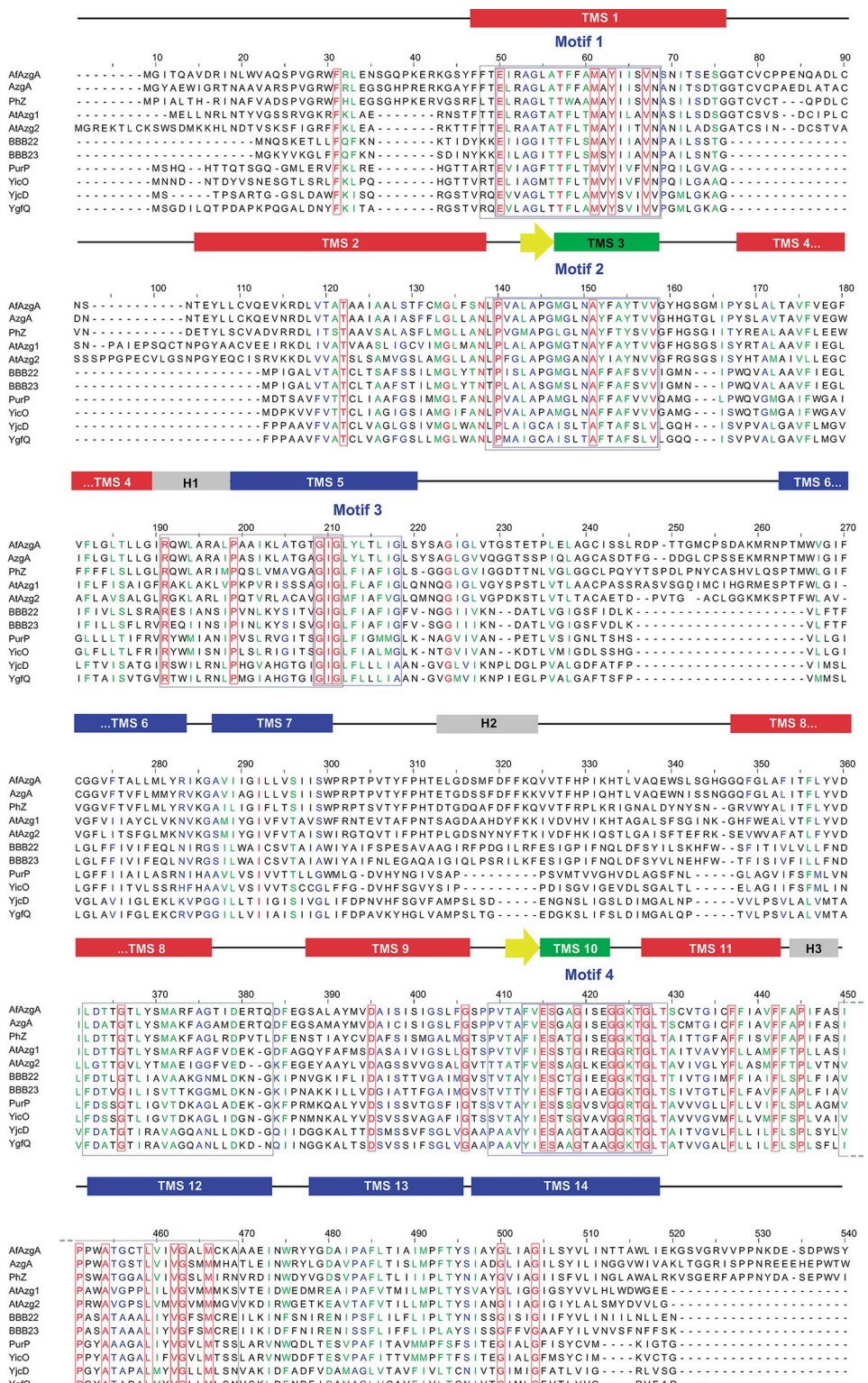

**Fig 1. Multiple sequence alignment of PhZ, and AzgA-like homologues with known function.** Aligned sequences include PhZ from *P. chrysosporium*; AzgA from *A. nidulans*; AfAzgA from *Aspergillus fumigatus*; AtAzg1 and AtAzg2 from *Arabidopsis thaliana*; PurP, YicO, YjcD, and YgfQ from *E. coli*; BBB22 and BBB23 from *Borrelia burgdorferi*. The region 541–610 of the alignment was omitted as it contained irrelevant data. Blue boxes indicate conserved motifs defined in this work, and grey boxes indicate those previously described [15]. Invariant residues in all analysed AzgA-

like proteins (including the 144 hypothetical proteins) are highlighted in red. The proposed secondary structure for PhZ according to the constructed three-dimensional model (see Fig 9) is schematically represented above the alignment. Transmembrane segments (TMS) are depicted as red, blue and green rectangles, β-sheets as yellow arrows, and internal helices as grey rectangles. TMSs of the gate domain are represented in blue, TMSs of the core domain are in red, and TMSs that contribute to the antiparallel β motif are depicted in green. The Accession numbers of the sequences used are listed in the S1 Appendix.

motifs 2 and 4 because they are part of the antiparallel β motif whose residues would intervene in the formation of the substrate binding site [7].

The residues to be analysed within these motifs were defined by looking for change-of-function mutations (i.e., substrate specificity, transport capacity) and minimising complete loss-of-function mutations. Thus, invariant residues, those conserved in plant and fungal sequences and those with higher variability (likely to have less impact on structure and function) were discarded. Those that differed between AzgA and PhZ were chosen among the remaining residues. In motif 2, residues L124, T131 and S133 of PhZ were replaced by the corresponding residue in AzgA (methionine, alanine and threonine respectively), to generate PhZ versions with changes: L124M, T131A and S133T. In motif 4, PhZ versions with the changes I388V, A391G and T392A were generated. In addition, seven randomly obtained mutations were analysed (see Materials and Methods): Y54G, V58A (motif 1); A128F, Y129D (motif 2); A148V, A418V and T429P (outside motifs) (Fig 2).

All versions of mutant PhZ fused to the gene encoding the green fluorescent protein (GFP) were expressed at the *azgA* locus of the *A. nidulans* ΔZAC strain lacking the major purine transporters (Δ*azgA*, Δ*uapA* and Δ*uapC*) (see Materials and Methods, S1 Table). In the same genetic context, the wild-type proteins PhZ and AzgA were expressed: PhZwt (Δ*ZAC::phZwt::*

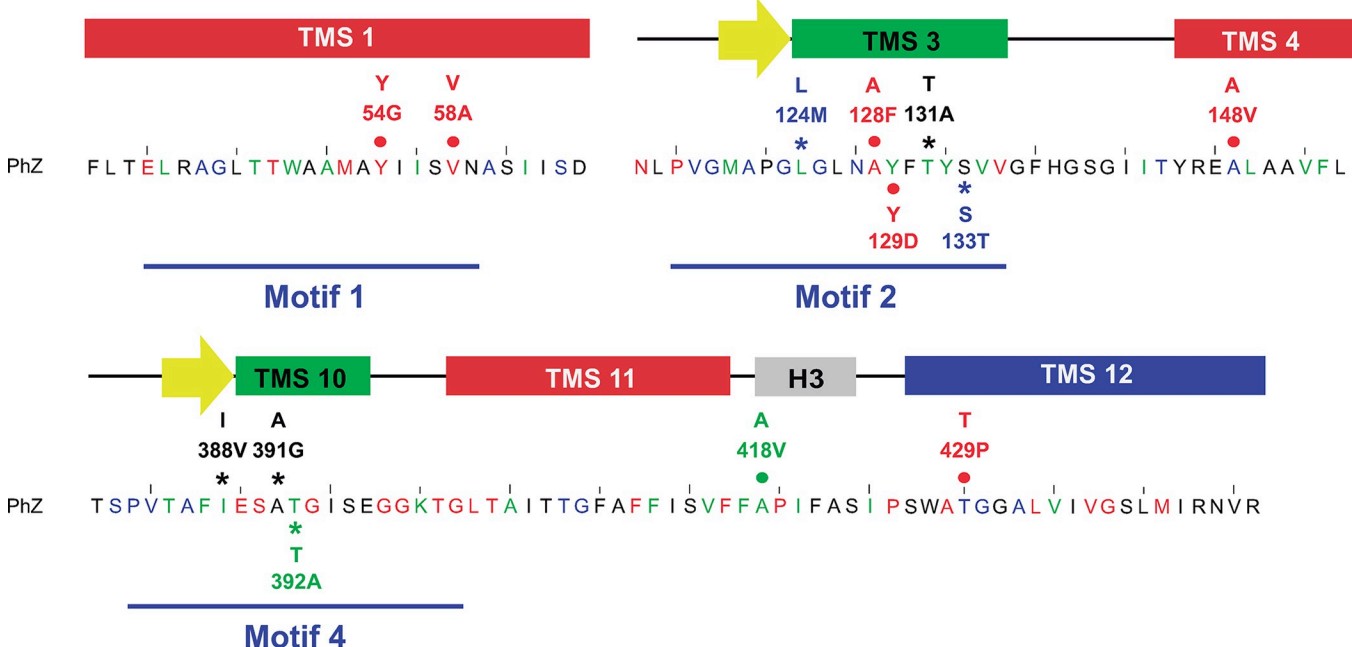

**Fig 2. Localisation of analysed PhZ residues.** Mutations resulting in loss of transporter function are marked in red, mutations causing decreased transport are in blue, mutations leading to increased transport are in green, and mutations with no observed effect are in black. The asterisk indicates rationally designed mutations, and the circle indicates mutations initially obtained through random mutagenesis and evaluated as double mutants. Subsequently, single mutants were constructed (see Materials and Methods).

*gfp*) and AzgA (*ΔZAC::azgA::gfp*) [17]. Thus, the constructed and control strains are isogenic, differing only in the version of the transporter under study.

It is important to highlight that GFP does not affect the activity of the transporter. Previously, the same growth phenotypes were observed in *A. nidulans* strains expressing wild-type PhZ with and without GFP: PhZwt and PhZwt12 (*ΔZAC::phZwt*) [17]. Here it was demonstrated by directly comparing hypoxanthine uptake (S2 Fig).

### Phenotypic and transport analysis of PhZ mutants

The resulting mutants were evaluated by growth tests in the presence of purines and analogues. Since the ΔZAC recipient strain lacks the main purine transporters and therefore cannot use purines as a nitrogen source, the growth of each constructed strain serves as a qualitative measure of the activity of the transporter (wild type or mutant) incorporated when purine is the only available nitrogen source [17]. If the mutation decreases transport, the growth of the corresponding colony is lower than that of the colony expressing the wild type, and improved growth indicates that the change causes an increase in transport. In contrast, the absence of growth in the presence of the toxic analogue 8-azaguanine shows that the protein under study transports this toxic analogue.

The 13 mutant strains' growth was evaluated using purines (hypoxanthine, adenine, guanine, xanthine, uric acid) and ammonium (a non-purine nitrogen source) as sole nitrogen source, and in the presence of the toxic analogue 8-azaguanine. For ease of reading, Fig 3 shows only the growth at 37˚C in the presence of ammonia, hypoxanthine, and 8-azaguanine (more informative data). The growth on other purines, oxypurinol, at 37˚C and 28˚C is presented in S3 Fig. The following control strains were included: ΔZAC, unable to grow in the presence of purines as a nitrogen source; PhZwt and AzgA both of which can grow using hypoxanthine, adenine, and guanine as nitrogen sources but not using uric acid or xanthine and is sensitive to 8-azaguanine [17]. Phenotypically, the differences between these strains are that hypoxanthine and 8-azaguanine are transported more efficiently through AzgA, reflected in the fact that the PhZwt strain grows less in hypoxanthine and more in 8-azaguanine than the AzgA strain (Fig 3).

Some mutants showed differences in growth on hypoxanthine and 8-azaguanine compared to the control PhZwt. Regarding growth on hypoxanthine, the same differences were observed when strains were grown on adenine and guanine (S3 Fig). Based on the growth phenotype on hypoxanthine/8-azaguanine, the evaluated mutants were grouped into four phenotypic groups: i) those that showed a phenotype equivalent to the ΔZAC strain: Y54G, V58A, A128F, Y129D, A148V, and T429P (phenotype -/+++); ii) mutants that showed an intermediate phenotype between the ΔZAC and PhZwt strains: L124M and S133T (phenotype -/++); iii) mutants that showed growth equivalent to the strain PhZwt: T131A, I388V, and A391G (phenotype +/+); and iv) those which can grow more than the control strain PhZwt on hypoxanthine and grow less on the toxic 8-azaguanine: T392A and A418V (phenotype ++/-) (Fig 3). These results suggest that the mutations evaluated in the first group cause the protein to lose its transport activity; the mutations L124M and S133T would reduce the transport activity; the mutations in the third group would not cause variations in the transport activity, and the mutations T392A and A418V could increase the capacity with which the substrates are transported (particularly 8-azaguanine).

Whether the mutations generate cryosensitive phenotypes was tested by observing growth at 28˚C in the presence of ammonium and hypoxanthine. No significant changes were obtained (S3 Fig). Finally, none of the mutants acquired the ability to transport new substrates such as those normally transported by UapA. That is, none of them acquired the ability to

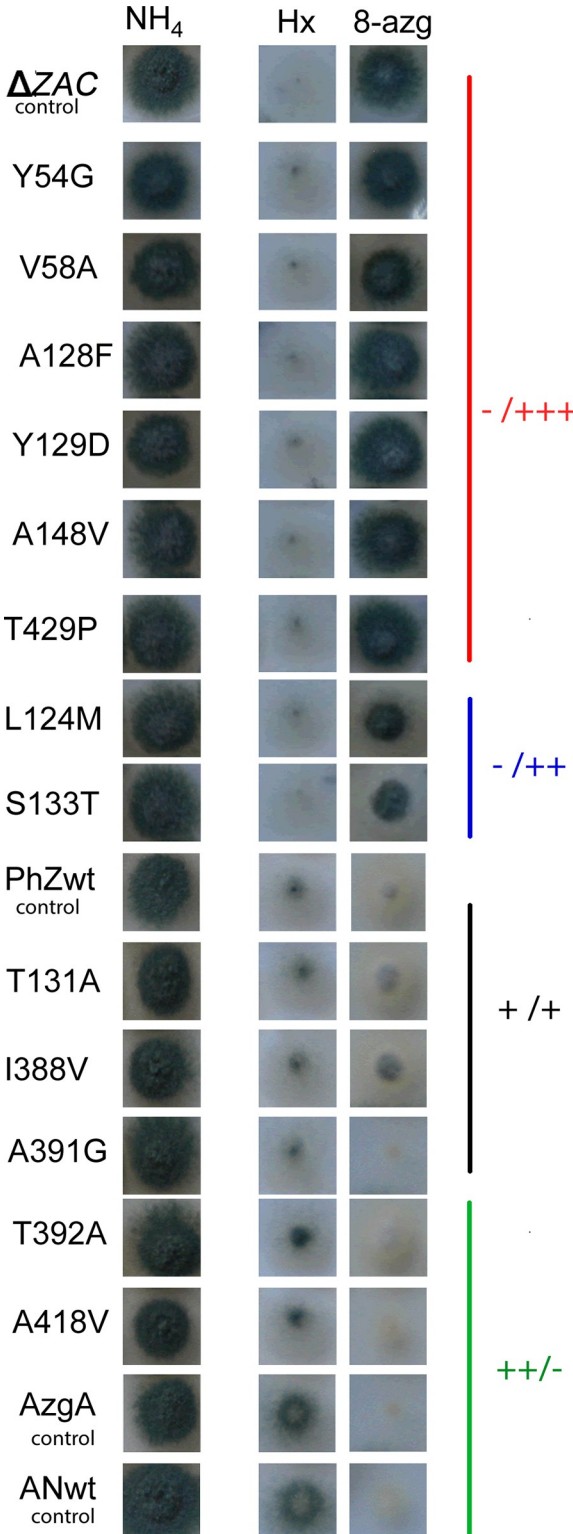

**Fig 3. Growth test analysis of PhZ mutants.** Strains were grown with ammonium tartrate 5 mM (NH$_4$), hypoxanthine 0.7 mM (Hx), and 8-azaguanine 0.7 mM + ammonium tartrate 0.8 mM (8-azg) at 37°C for 48 hours. Control strains included: ΔZAC, PhZwt and AzgA. Evaluated mutants are grouped into four categories based on their growth phenotype on hypoxanthine/8-azaguanine: -/+++ (red), -/++ (blue), +/+ (black), and ++/- (green).

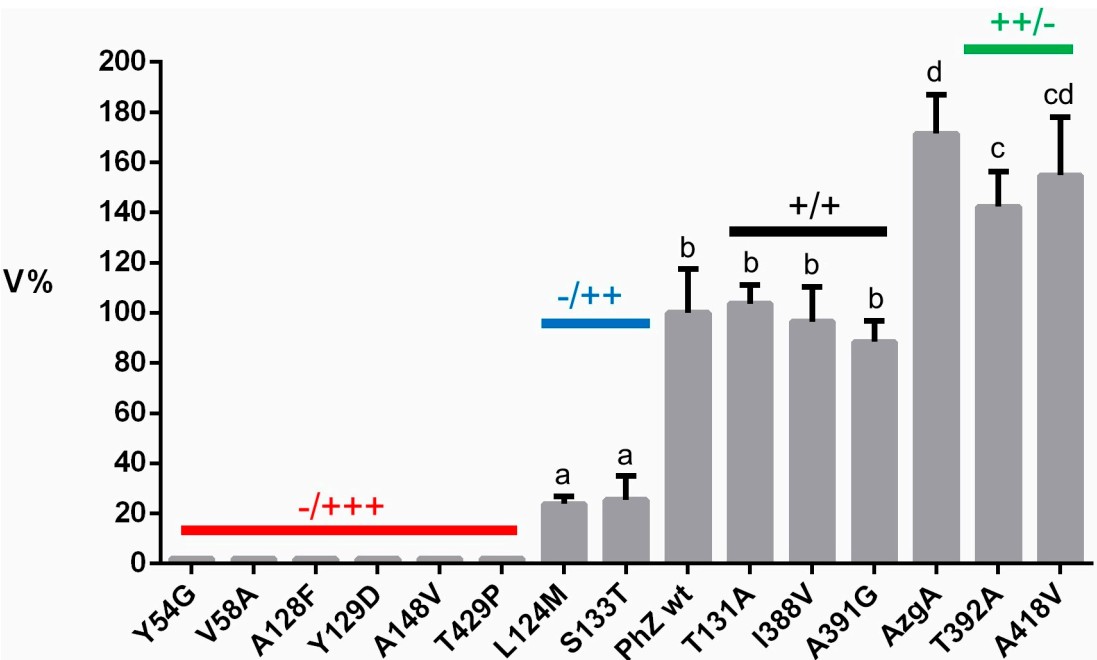

**Fig 4. Initial transport rate of [³H]-hypoxanthine of PhZ mutants (V%).** AzgA, PhZwt and mutant strains were included. 100% is considered the transport rate of the PhZwt strain. Mutants with measurements equivalent to those obtained with the ΔZAC strain are marked in red; mutants with decreased hypoxanthine uptake, equivalent to or higher than PhZwt, are indicated in blue, black or green, respectively. Given the low V% value obtained with strains Y54G, V58A, A128F, Y129D, A148V and T429P (<2%), an arbitrary V% value of 2 was assigned for presentation purposes. Different letters (a, b, c, d) represent significant differences at p < 0.05 probability level, according to ANOVA and Tukey´s test.

grow on uric acid or xanthine and all remain resistant to the toxic analogue oxypurinol (S3 Fig).

As a quantitative measure of transport, the initial transport rate (V) of each strain was determined by uptake assays performed with radiolabelled hypoxanthine ([³H]-hypoxanthine) (see Materials and Methods). The initial rate of the mutants is expressed as a percentage (V%) concerning the PhZwt strain to which the value 100% was assigned. The results of this analysis (Fig 4) are consistent with those obtained by growth test analysis. The first group, composed of Y54G, V58A, A128F, Y129D, A148V, and T429P, shows a negligible hypoxanthine transport rate. In the second group, mutants L124M and S133T show a decrease in transport of approximately 75%. Mutants T131A, I388V, and A391G showed no significant difference from the wild-type transporter, and T392A and A418V showed a hypoxanthine uptake rate higher than that of PhZwt and equivalent to that of the AzgA transporter.

All mutants that showed differences in transport concerning the wild-type PhZ transporter and detectable [³H]-hypoxanthine transport activity (L124M, S133T, T392A, and A418V) were subjected to hypoxanthine $K_{m/i}$ determination (Fig 5). None of them showed significant variations concerning the $K_{m/i}$ value of the wild-type PhZ transporter. The only mutant that showed a slightly lower $K_{m/i}$ value was L124M, indicating that while this mutant has reduced initial hypoxanthine uptake its affinity for hypoxanthine is slightly increased concerning the wild-type PhZ transporter.

Furthermore, to investigate the possible impact of these four mutations on the profile of transported substrates, the ability of other physiological substrates (i.e., adenine, guanine and 8-azaguanine) to compete with [³H]-hypoxanthine transport was evaluated. Although it is an indirect method (see Materials and Methods), it has the advantage of evaluating numerous

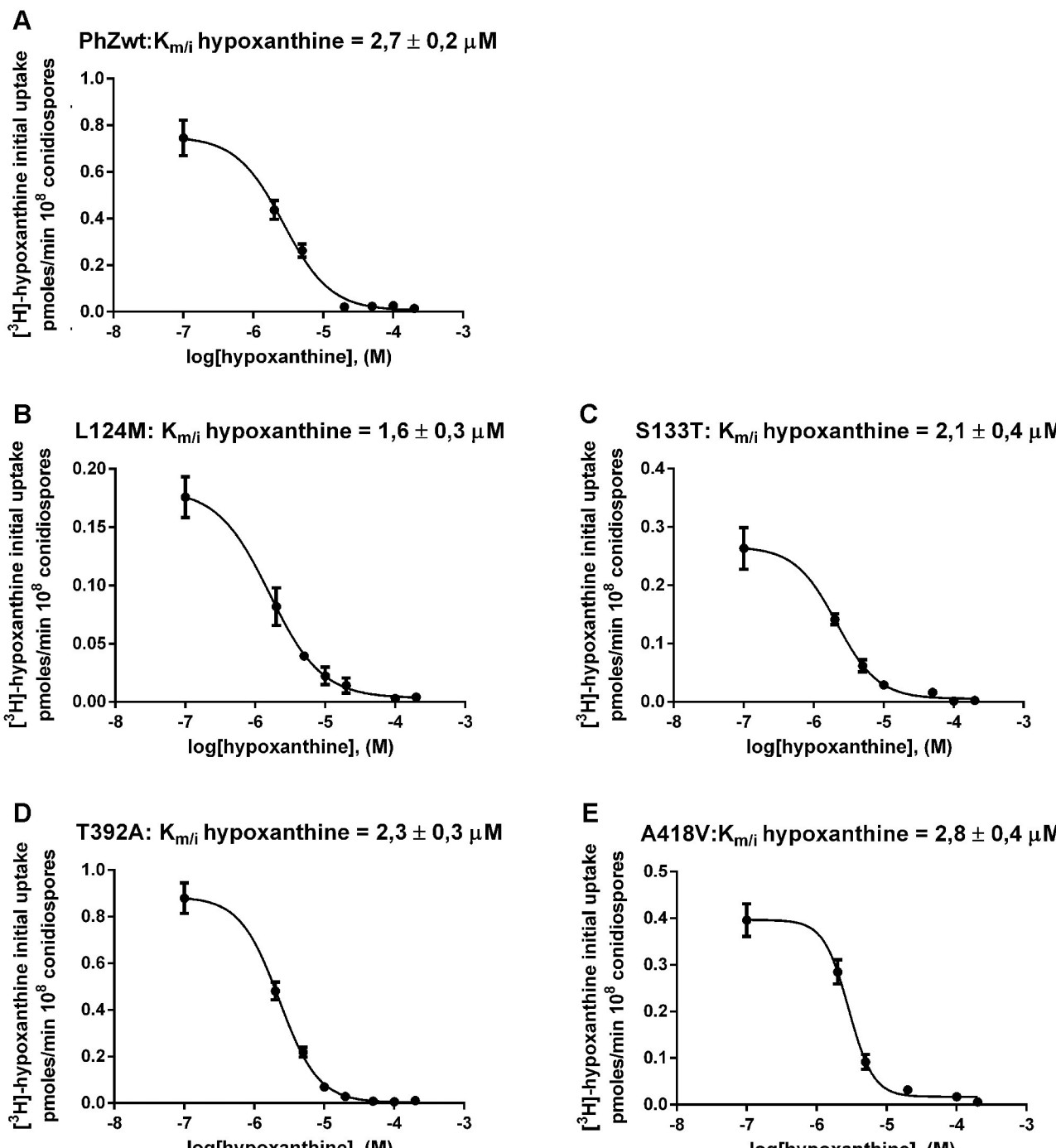

**Fig 5. Kinetic characterisation of PhZ mutants.** Dose-response curves for hypoxanthine are shown. The uptake of [3H]-hypoxanthine was measured in the presence of increasing concentrations of unlabelled hypoxanthine (0.1–1 mM). The $K_{m/i}$ value was determined as described in Materials and Methods. The standard deviation was <20% in all cases. (A) PhZwt; (B-E) L124M, S133T, T392A and A418V mutants respectively.

substrates using a single labelled substrate, provided that the protein has detectable transport activity for that labelled substrate. Generally, under the conditions of these assays with substrates that are transported at the level of the labelled substrate, a percentage of uptake of ~1–5% is obtained [18]. The four mutants showed significative differences from the wild-type PhZ

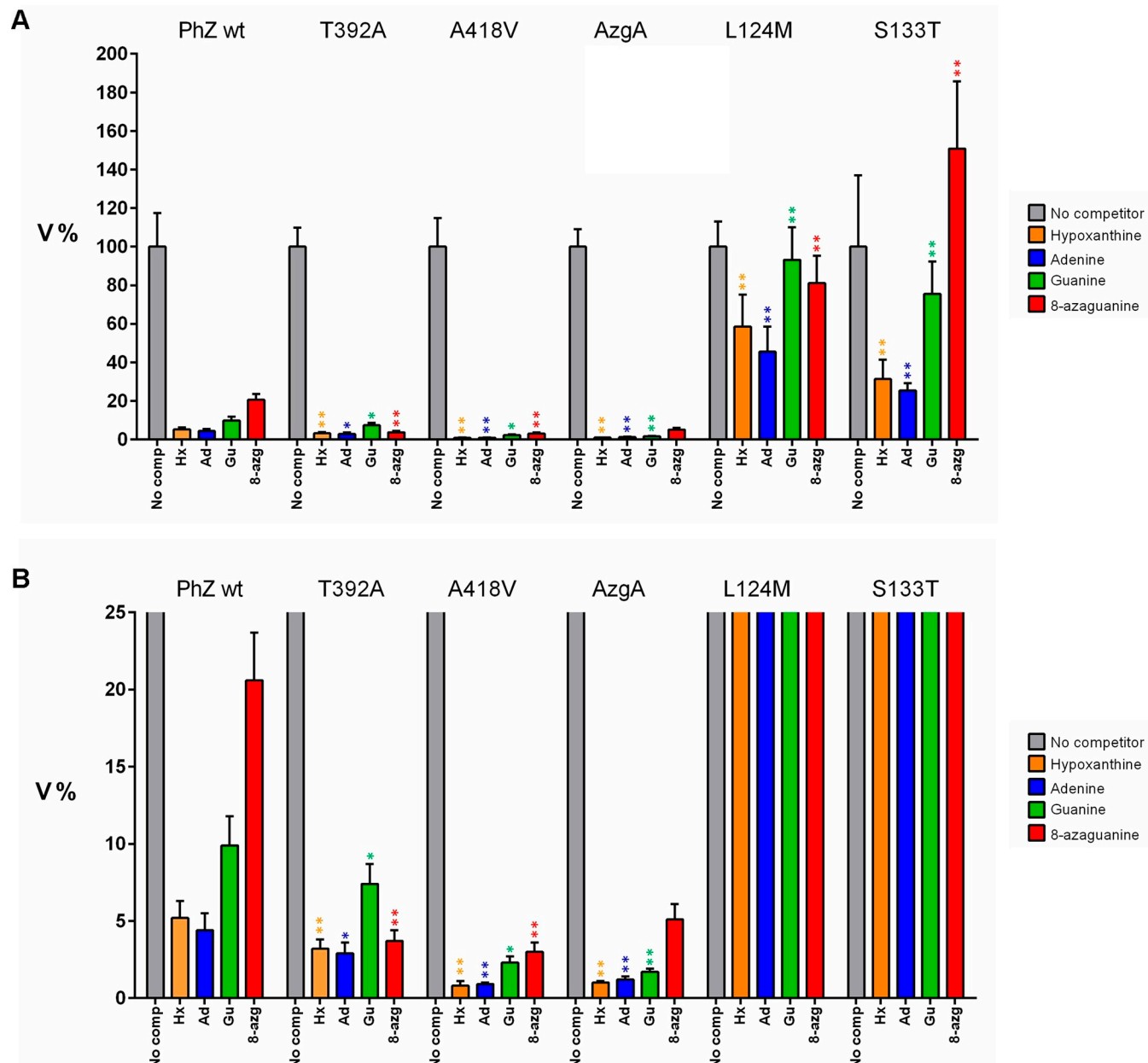

**Fig 6. Transport assays of [³H]-hypoxanthine in the presence of other unlabelled substrates.** (A) Graphs show the percentage of initial transport velocity of [³H]-hypoxanthine in the presence of 1 mM of the indicated competitor substrates. 100% was considered the transport rate obtained without competitor substrate, and the initial uptake rate for each strain (Fig 4) was used to normalise the results. Unlabelled hypoxanthine competition was included as a control, showing maximum inhibition percentage (~95–99%). (B) X-axis (V%) enlargement for clarity. Asterisks represent significant differences with the respective condition in PhZwt at $p < 0.05$ (*) or $p < 0.01$ (**) probability level, according to ANOVA and Dunn's test with Bonferroni and Holmes correction.

with all substrates tested (Fig 6). L124M and S133T, show less [³H]-hypoxanthine inhibition regarding PhZwt with all purines, meanwhile the T392A and A418V mutants exhibited higher levels of inhibition, with a closer inhibition profile to that observed for the AzgA protein (Fig 6B). Hence, these substitutions lead to increased competition from physiological substrates for binding to PhZ.

### Expression levels and plasma membrane localization of mutant versions of PhZ

The observed differences in transport rates and phenotypic analysis could imply problems with traffic towards the plasma membrane, an increase in protein turnover rate (due to protein instability at the plasma membrane) or a defect in the synthesis of mutant versions of PhZ. To rule out these possibilities, subcellular localization analyses were carried out by microscopic observation of green fluorescent protein, and protein synthesis was analysed by western blot with anti-GFP antibody (Figs 7 and 8).

It is important to note that the wild-type version of PhZ, in addition to its localization in the plasma membrane and vacuoles (as a result of normal protein turnover), can also be observed in cytoplasmic rings, which probably correspond to the endoplasmic reticulum (ER)

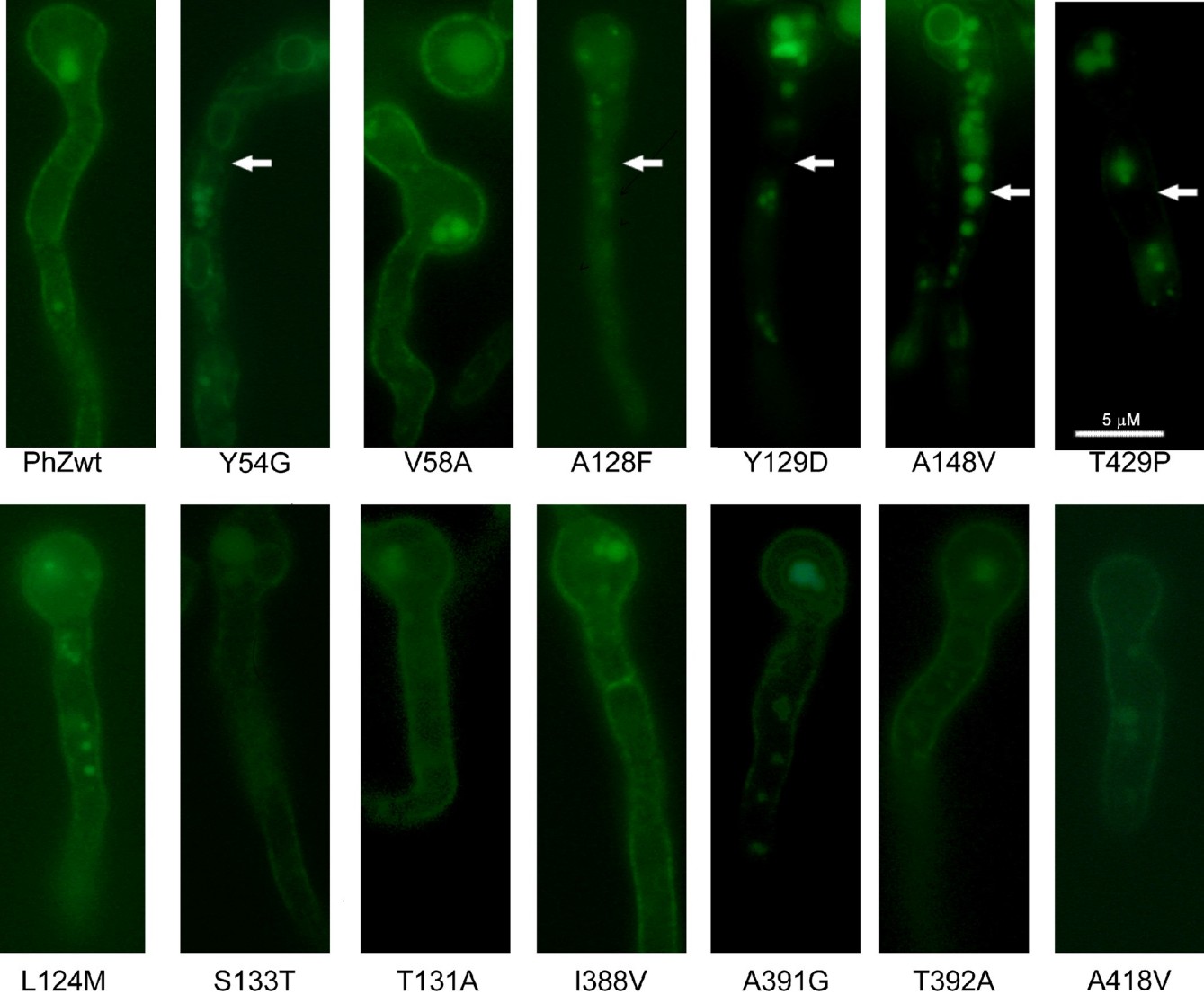

**Fig 7. Subcellular localisation of PhZ mutants.** Epifluorescence microscopy of strains expressing PhZ, wild-type and mutants, fused to green fluorescent protein (GFP). A representative image of the behaviour observed in 10–15 analysed fields is shown for each strain. Arrows indicate the absence of fluorescence at the membrane level. In the other mutants as in the PhZwt control, fluorescence is observed in the membrane, vacuoles, and cytoplasmic rings corresponding to the ER (pattern previously determined in [17]).

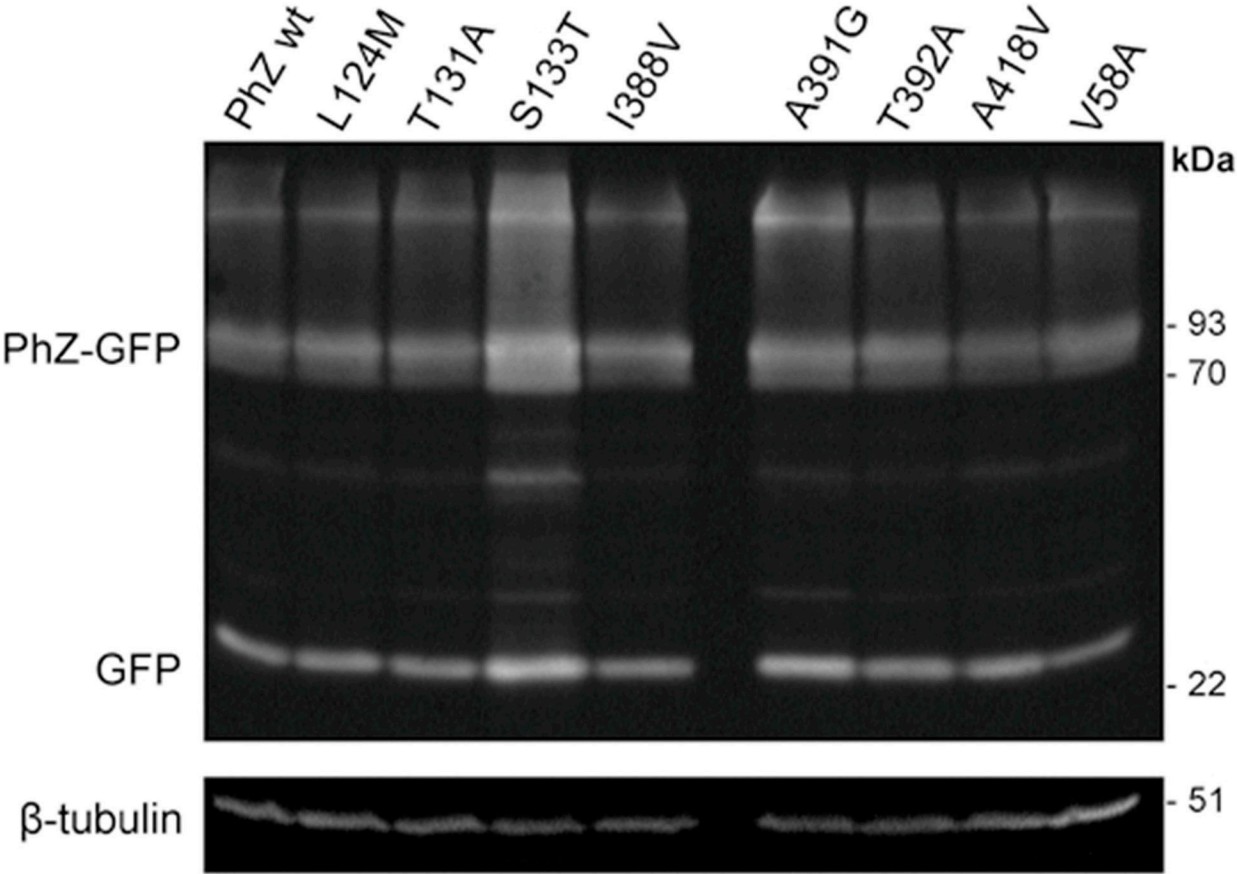

**Fig 8. Western blot of total protein extracts.** Total proteins from strains expressing PhZ, wild-type and mutants, fused to green fluorescent protein (GFP) were analysed by probing with monoclonal antibodies against GFP and tubulin (loading control). The band corresponding to intact PhZ-GFP (low mobility) and free GFP (high mobility) are indicated. Bands with lower mobility than PhZ-GFP may correspond to PhZ-GFP fusion oligomers or protein aggregates (due to their high hydrophobicity). Bands observed between the indicated PhZ-GFP and free GFP bands may correspond to degradation intermediates. The original image of the Western blot is provided in S1 Raw image.

[17]. Fluorescence microscopy analyses (Fig 7) revealed that most mutants show a similar localization to the PhZ-GFP wild-type version. Of the six mutants that showed a transport-deficient phenotype (equivalent to the ΔZAC strain), only V58A can reach the plasma membrane. In the other five mutants (Y54G, A128F, Y129D, A148V, and T429P), fluorescence is mainly localized in vacuoles. In the case of Y54G, it is also observed that a large part of the protein is retained in perinuclear rings that probably correspond to the endoplasmic reticulum. These results indicate that the replaced residue in these five mutants probably affects folding, being in some cases retained by the quality control mechanisms operating at the ER, or rapidly degraded, preventing the protein localisation at the plasma membrane. Other residues' replacement does not significantly affect the plasma membrane localisation. Only in mutant S133T, it is observed that the amount of protein associated with vacuoles and ER slightly increases compared to that located in the plasma membrane.

To investigate whether the observed phenotype in mutants that showed similar localization to wild-type PhZ (L124M, T131A, S133T, I388V, A391G, T392A, A418V, and V58A) was due to changes in the amount of expressed protein and/or its stability, their protein synthesis was evaluated by western blot. It is important to mention that in this technique, it is common to observe a band corresponding to the protein-GFP fusion and another corresponding to free

GFP, given the resistance of the latter to proteolysis [34]. This resistance allows free GFP to be used as an indirect standard to assess the vacuolar degradation of membrane proteins fused to it [35, 36]. The results show that in all mutants, the protein level is similar to that of the wild-type PhZ (Fig 8). Regarding the protein stability of the different mutants analysed, the free GFP signal is significantly more intense in S133T and A391G. This also applies to the S133T-GFP fusion. Therefore, we considered that these mutants would present differences in stability compared to wild-type PhZ.

## PhZ structural model

The results of the mutational analysis were correlated with the PhZ structure using the 3D model constructed for this purpose (see Materials and Methods). The resulting model (Fig 9A) proposes a 3D structure of PhZ consisting of 14 TMS, three internal helices, and an antiparallel β-motif formed by a β-sheet next to TMS3 and another next to TMS10. The model corresponds to a conformation occluded towards the cytoplasm. The protein structure is topologically divided into two domains, core and gate domains, formed by two inverted repeats. The core domain consists of 8 TMS, comprising TMS1-4 and TMS8-11, while the gate domain consists of 6 TMS, formed by TMS5-7 and TMS12-14. Through analysis of the electrostatic profile, it was observed that the distribution of ionized residues on the protein surface is quite reasonable: most are placed on the cytoplasmic or extracellular side or along the protein pore in its interior. On the outer side, a slightly negative distribution is observed, while the cytoplasmic side appears positively charged, and the intermediate region is neutral, consistent with the position in the lipid bilayer. A strongly negative "patch" can be observed in the interior region comprising TMS8, TMS5, and internal helix 1, which would be part of the protein pore (Fig 9A and S4 Fig). The 3D structure of the AzgA-like transporter from *Arabidopsis thaliana* recently published supports the findings of our model, with an RMSD of 1.53 Å between them [16]. Finally, the stereochemical evaluation (PROCHECK) of most residues falling within favourable regions of the Ramachandran plot and G-factors close to zero (-0.07) indicated good overall stereochemical quality (S5 Fig).

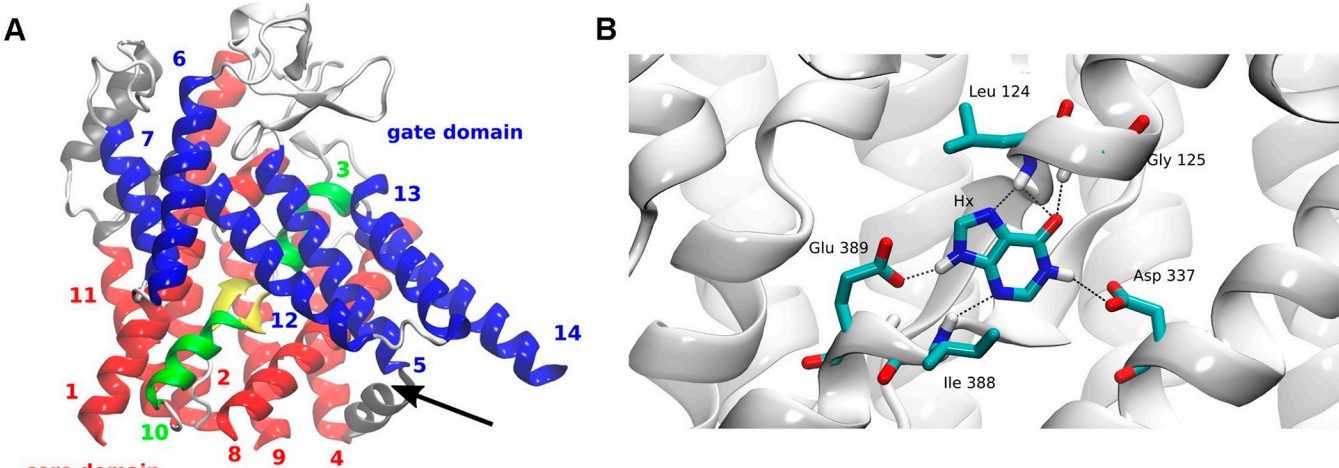

**Fig 9. Theoretical model of PhZ.** (A) Schematic representation of the 3D model. Transmembrane segments (TMSs) are displayed as ribbons with numbering indicated. TMSs forming the core domain are shown in red, while those of the gate domain are depicted in blue. β-sheets located in the active site and the TMSs that contribute to the antiparallel β motif (in yellow) are represented in green. Internal loops and helices are represented in grey. The black arrow indicates the protein pore in the inner region between TMS8, TMS5 and internal helix 1. (B) The residues interacting with the substrate (hypoxanthine) through hydrogen bonds are marked in colours. Dashed lines indicate hydrogen bonds. For more details regarding substrate docking, refer to Fig 10.

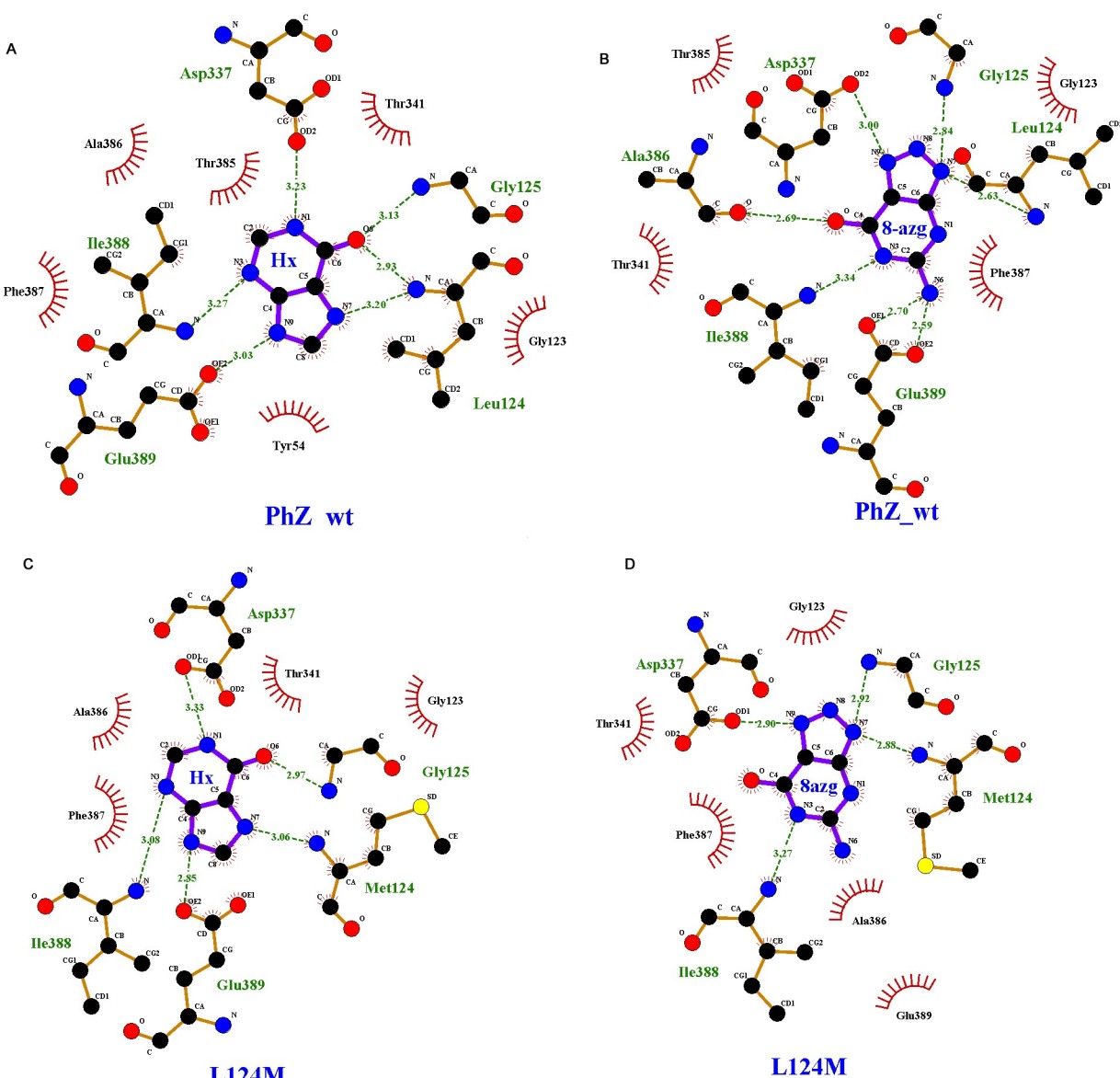

**Fig 10. Substrate docking.** 2D schematic representations (LIGPLOT program) illustrating the interactions of the best conformation obtained from molecular docking of: PhZwt-Hypoxanthine (A), PhZwt-8-azaguanine (B), L124M-Hypoxanthine (C), L124M-8-azaguanine (D). Hydrogen bonds and their distances are shown as green dashed lines; other interactions are depicted as red wavy lines.

Regarding the localisation of the thirteen residues analysed in this study in the 3D structure, according to the proposed model: two are located in TMS1 (Y54 and V58); five in TMS3 (L124, A128, Y129, T131 and S133); one is in TMS4 (A148); three in TMS10 (I388, A391 and, T392), one in the internal helix between TMS11 and TMS12 (A418), and one in the loop between TMS11 and TMS12 (T429). Thus, 8 of the 13 residues are part of the antiparallel β motif. Among them, L124 and I388 are two residues that, according to the proposed model, would interact with 8-azaguanine and hypoxanthine (Figs 9B and 10).

Regarding the latter, the substrates hypoxanthine and 8-azaguanine were docked into the modelled structure of PhZ (see Materials and Methods) to explore possible protein-substrate interactions. We found that most residues interacting with both substrates are analogues to the

ones reported before for AzgA (PhZ residues G125, D337, I388 and E389, correspond to residues G129, D342, V393, and E394, in AzgA, respectively) [7]. Notably, L124 (M128 in AzgA) emerges as a new interactor for substrates tested (Fig 10). In addition, non-covalent interactions also serve to support the proposed binding site, π-π stacking interactions with F387 would contribute to the stabilization of the binding site (Fig 10), similar to what has been observed with F335 in AzgA [7], It is postulated that the bulky chains of these aromatic residues would isolate the substrate in the occluded conformation, and its movement in the open conformation would release it [7, 10, 15, 37].

These non-covalent interactions also serve to support the proposed binding site, as all similar transporters that have been modelled have one or two phenylalanine residues forming this type of stabilizing interaction: F335 in AzgA; F144 and F406 in UapA [7, 10].

According to the docking results, the side chains of residues D337 and E389 and the amino group of I388, L124 and G125 would interact with N1 (H), N3 (H), and C6 = O (pyrimidine ring) and N9 (H) (imidazole) of the hypoxanthine substrate. When the substrate is 8-azaguanine, A386 incorporates into the binding site by interacting with the oxygen of its carbonyl group with C6 = O of the pyrimidine ring, leading to a reorganisation of the interactions involving D337, E389, I388, L124 and G125 (Figs 9 and 10). As an interesting case, the L124M mutant was also analysed by docking (Fig 10). Docking with hypoxanthine revealed that while in the wild-type, L124 interacts with C6 = O of the pyrimidine ring and N7 (H) of imidazole, M124 only interacts with N7 (H). When the substrate is 8-azaguanine, the substitution of leucine for methionine at this position results in a loss of interactions with N337 and E389.

## Discussion

Table 1 summarises the results obtained. Thirteen mutations were analysed in this work, three of which (T131A, I388V and A391G) showed behaviour equivalent to the wild-type transporter. These mutations do not affect growth, trafficking to the plasma membrane nor transport activity (Figs 3, 4 and 7). Regarding protein synthesis and turnover, only mutant A391G seems to show a slightly increased stability (i.e., the band corresponding to this PhZ-GFP version is slightly more intense than that of the wild-type transporter, Fig 8), but this does not seem to affect the transporter's activity. According to the proposed model (Figs 9 and 10), the three residues are located on the periphery of the proposed binding site, and I388 may interact with the substrate. In AzgA, the analogue residue to I388 is V393, whose amino group also interacts with hypoxanthine in the proposed model for this *A. nidulans* protein [7]. In an analogous position, among other AzgA-like proteins (S1 Fig), some have isoleucine as PhZ, and others have valine as AzgA. The results are consistent with the proposed model since the I388V change in PhZ does not produce significant variations compared to the wild-type protein.

Ten of the analysed mutants showed differences in transport compared to wild-type PhZ (Figs 3 and 4), either because they caused a complete loss of function (Y54G, V58A, A128F, Y129D, A148V, and T429P), a decrease (L124M and S133T), or an increase in transport (T392A and A418V).

The complete loss of function observed in mutants Y54G, A128F, Y129D, A148V, and T429P can be explained by the fact that they do not reach the plasma membrane, probably due to misfolding (Fig 7). Most of these residues are located in regions to which functional importance has already been assigned [7]: TMS1 (Y54), TMS3 (A128 and Y129), and TMS10 (T392), the latter two forming part of the anti-parallel β-motif. Analysis of UapA, AzgA, and other related transporters suggests that residues in these TMSs perform functions related to substrate trajectory, transporter specificity, or the formation of critical polar interactions [10, 15, 18, 37].

**Table 1. Functional analysis summary of PhZ mutants.**

| Mutation | Location in protein* | Hypoxanthine, adenine or guanine Growth | 8-azaguanine Growth | V(%) | Subcellular localisation equivalent to PhZ wt |
|---|---|---|---|---|---|
| Y54G | TMS 1 | − | +++ | < 2 | no |
| V58A | TMS 1 | − | +++ | < 2 | yes |
| A128F | TMS 3 | − | +++ | < 2 | no |
| Y129D | TMS 3 | − | +++ | < 2 | no |
| A148V | TMS 4 | − | +++ | < 2 | no |
| T429P | TMS11-12 | − | +++ | < 2 | no |
| L124M | TMS 3 | − | ++ | 23±3 | yes |
| S133T | TMS 3 | − | ++ | 21±2 | yes** |
| T131A | TMS 3 | + | + | 103±7 | yes |
| I388V | TMS 10 | + | + | 96±14 | yes |
| A391G | TMS 10 | + | + | 88±8 | yes |
| T392A | TMS 10 | ++ | − | 142±14 | yes |
| A418V | H3 | ++ | − | 155±23 | yes |
| **Control** | | | | | |
| PhZwt | | + | + | 100±17 | n.a |
| AzgA | | ++ | − | 155±23 | n.a |
| ΔZAC | | − | +++ | < 2 | n.a |

*According to the constructed three-dimensional models. V (%) represents the percentage of initial hypoxanthine uptake by taking V% of PhZwt as 100. n.a.: not applicable.

** indicates that a greater amount of protein associated with vacuoles is observed. H3 corresponds to the internal helix located between TMS11 and TMS12. Growth phenotype: equivalent to PhZwt (+); higher than PhZ (++, +++); lower than PhZ (-).

The exceptions are A148V and T429P located, according to the model, in TMS4 and between TMS11 and TMS12. For this region, no previous reports were found assigning a specific role. These results allow us to conclude that Y54 (TMS1), A128 (TMS3), Y129 (TMS3), and A148 (TMS4) could be important for maintaining the transporter's architecture. In the case of T429, substitution by proline induces a structural change. Proline is considered a helix terminator due to the lack of rotation of its Cα atom, and when integrated into a pyrrolidine ring, it hinders the formation of hydrogen bonds, disrupting the helix stability. Regarding mutant V58A, this is the only complete loss of function mutant that reaches the plasma membrane and no modifications in traffic or differences in protein synthesis or turnover rate are observed compared to wild-type PhZ (Figs 7 and 8). Therefore, the differences observed in [³H]-hypoxanthine uptake and growth tests (Figs 3 and 4) reflect a true decrease in transport activity, and it can be concluded that V58 (which localises on TMS1) is key to PhZ functionality. This residue is located within one of the previously defined AzgA-like motifs (motif 1, Figs 1 and 2) and is one of the absolutely conserved residues. According to our model, it would be at the periphery of the main binding site, less than 10 Å away from the substrate. The functional importance of the TMS1 residues has already been demonstrated through the analysis of other invariant residues of motif 1 [7, 15].

Mutations L124M and S133T showed growth impairment on PhZ physiological substrates (Fig 3 and S3 Fig) in spite of reaching the plasma membrane and not showing differences in protein synthesis and turnover compared to wild-type PhZ (Figs 7 and 8). [³H]-hypoxanthine uptake assays revealed a ~75% transport decrease of this substrate (Fig 4), which explains the observed growth phenotype. Both mutations showed a similar profile regarding [³H]-hypoxanthine transport competition: decreased inhibition by excess unlabelled physiological substrates (i.e., adenine, guanine, hypoxanthine and 8-azaguanine), which correlates with the

observed growth impairment. (Fig 6). As L124M and S133T mutations affect binding and or substrate translocation, we conclude that L124 and S133 are critical residues for PhZ activity.

The substitution L124M has a slightly increased apparent affinity (decreased $K_{m/i}$, Fig 5). While this may seem contradictory at first, in AzgA, it has been shown that mutations that increase affinity cause the substrate to be "partially trapped," affecting transport capacity [7]. This result is consistent with the interaction model proposed for PhZ-hypoxanthine and PhZ-8-azaguanine where L124 would directly interact with the substrate (Fig 10). In the wild-type version of the transporter, hypoxanthine and 8-azaguanine establish six and seven hydrogen bonds in the putative binding site, respectively (Fig 10A). In the L124M mutant, hypoxanthine and 8-azaguanine establish five and four hydrogen bonds, respectively (Fig 10B). This reduction in hydrogen bonds may explain why the L124M mutant exhibits an inefficient transport of the substrates mentioned above. Since the protein is a transporter, the reduced number of interactions may result in the substrate being partially trapped and, for instance, not spending enough time in the active site, which could impair its proper translocation. Also, this loss of interactions may preclude the correct positioning of this substrate in the putative binding site. These results underscore the critical role of residue L124 in the PhZ substrate binding site, identifying it as an additional key determinant in the transport mechanism of AzgA-like transporters. It would be valuable to investigate the role of analogous residues in other members of the AzgA-like family.

The S133T substitution does not modify its binding affinity for hypoxanthine (Fig 5). This would indicate that this mutation does not affect the folding or substrate binding affinity. Notwithstanding, western blot results showed that the band corresponding to this mutant (PhZ-GFP) is more intense than that of the wild-type transporter, indicating a higher level of protein expression (Fig 8). This could be due to an increase in the intrinsic stability of the mutant protein (i.e., a higher structural rigidity) that could be affecting its transport activity, which would explain the decrease in the hypoxanthine uptake rate (Fig 4). This increase in structural rigidity/stability could also result in an increase resistance to proteolysis, which might explain why a greater amount of mutant protein can be observed in vacuoles (Fig 7).

Finally, mutants T392A and A418V showed similar membrane localisation and protein synthesis and turnover compared to wild-type PhZ (Figs 7 and 8), while exhibiting better growth on adenine, guanine and hypoxanthine, and an increased sensitivity to 8-azaguanine (Fig 3 and S3 Fig). This correlates with the increase of initial uptake of [3H]-hypoxanthine, without modifying the binding affinity for this purine (Figs 4 and 5), and the fact the unlabelled PhZ physiological substrates exhibited a higher [3H]-hypoxanthine transport competition (particularly in mutant A418V) (Fig 6). According to the proposed model, neither of these residues would be part of the main binding site. This explains why the mutations would not affect the binding affinity, but the model does not explain the increase in the uptake rate. T392 is located in the TMS10 at the periphery of the main binding site, and although it is not part of the site, it is less than 10 Å from the substrate. In UapA, residues in the TMS10 (G411, T416, R417) that do not belong to the substrate binding site but are located in its periphery have been assigned a role related to the substrate trajectory [37]. A418 is the first residue of the internal helix located between TMS11 and TMS12. The residue A441 of UapA, which is located in this internal helix of the *A. nidulans* protein far from the substrate binding site (like A418 of PhZ), has been attributed a role related to substrate specificity, as mutations in this residue (A441V) result in a transporter with differences in specificity. Strains expressing UapA with A441V do not affect the transport of their physiological substrates and can grow on hypoxanthine (non-physiological substrate), albeit with low affinity for this purine. This suggests that the ability to transport non-physiological substrates in this mutant is due to an increase in transport capacity per se, rather than an increase in binding affinities [13, 37]. The

experimental evidence obtained here indicates that A418 of PhZ plays a key role in transport, and this is the first work that identifies critical residues for transport in the internal helix between TMS11-TMS12 in an AzgA-like transporter. While our experimental evidence clearly establishes the pivotal role of A418 in PhZ transport, it is noteworthy that our findings align with the previously published by the Diallinas group in UapA, underscoring the challenges in predicting the overall function of elevator-type transporters based solely on the character of amino acid residues directly engaging with substrates [37]. Our study offers empirical support for the notion that also in AzgA-like proteins seemingly unrelated residues, such as A418 in PhZ, can exert significant influence over the transport process, thus enhancing our comprehension of the intricate interplay within these transporter systems.

## Supporting information

**S1 Fig. Amino acids present in the four AzgA-like motifs defined in this study.** The analysis was based on the alignment obtained through ClustalW of AzgA-like proteins with known functions and 144 hypothetical ones (S1 Appendix). The numbering of the marked amino acids corresponds to the PhZ sequence. X represents any amino acid.
(TIF)

**S2 Fig. GFP incidence in initial [$^3$H]-hypoxanthine transport rate (V%).** Strains PhZwt and PhZwt12 expressing the GFP- and non-GFP transporter, respectively, are included. 100% is considered the transport rate of the PhZwt strain. Results are the average of two independent experiments each measured in triplicate. Identical letters (a) indicate no significant differences were found according to ANOVA and Tukey's test (p = 0.616).
(TIF)

**S3 Fig. Growth analysis of PhZ mutants.** This figure complements Fig 3. Left panel: growth in the presence of ammonium L(+) tartrate (NH$_4$), adenine (Ad), guanine (Gu), uric acid (UA), xanthine (X), and oxypurinol + 10 mM sodium nitrate (Oxy) at 37˚C for 48 hours. Right panel: growth on ammonium (NH$_4$) and hypoxanthine (Hx) at 28˚C for 4 days. Control strains: ΔZAC and PhZwt.
(TIF)

**S4 Fig. Electrostatic profile of PhZ.** The charge is indicated by the colour bar in the figure. A highly negatively charged red area, which would be part of the protein's pore, in the inner area between TMS8, TMS5, and internal helix 1, is indicated with an arrow.
(TIF)

**S5 Fig. Ramachandran plot.** The plot shows the distribution of φ (phi) and ψ (psi) angles for the amino acid residues in PhZ wt. Analysis performed using PROCHECK.
(TIF)

**S1 Table. *Aspergillus nidulans* strains used in this study.**
(DOCX)

**S2 Table. Primers used in this study.**
(DOCX)

**S1 Appendix. Accession number of the sequences used in this study.**
(DOCX)

**S1 Raw image.**
(TIF)

## Author Contributions

**Conceptualization:** Mariana Barraco-Vega, Manuel Sanguinetti, Gianna Cecchetto.

**Formal analysis:** Mariana Barraco-Vega.

**Funding acquisition:** Mariana Barraco-Vega.

**Investigation:** Mariana Barraco-Vega, Manuel Sanguinetti, Gabriela da Rosa.

**Methodology:** Mariana Barraco-Vega, Manuel Sanguinetti, Gabriela da Rosa.

**Project administration:** Mariana Barraco-Vega.

**Resources:** Mariana Barraco-Vega, Manuel Sanguinetti.

**Supervision:** Mariana Barraco-Vega, Manuel Sanguinetti, Gianna Cecchetto.

**Visualization:** Mariana Barraco-Vega, Manuel Sanguinetti.

**Writing – original draft:** Mariana Barraco-Vega, Manuel Sanguinetti, Gabriela da Rosa, Gianna Cecchetto.

**Writing – review & editing:** Mariana Barraco-Vega, Manuel Sanguinetti, Gabriela da Rosa, Gianna Cecchetto.

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
