## [Decision Letter · Decision Letter 0]

23 Aug 2024

PONE-D-24-30056Mutational analysis of Phanerochaete chrysporium´s purine transporterPLOS ONE

Dear Dr. Sanguinetti,

Thank you for submitting your manuscript to PLOS ONE. After careful consideration, we feel that it has merit but does not fully meet PLOS ONE’s publication criteria as it currently stands. Therefore, we invite you to submit a revised version of the manuscript that addresses the points raised during the review process.

**While the reviewers were positive about your manuscript, they also raised some concerns and suggestions that can improve its scientific impact. During your revision, please, consider all the Reviewer’s comments, but, please, pay particular attention to those indicating to:****Reorganize the introduction section and concisely describe the main objective and contribution of the work.****Avoid repeating the experimental approaches in the Results section and rather focus in describing the results obtained.****Some caption figures need to be rewritten to summarize the information  (Figure 1) and/or clarifying specific elements (Figs. 3 and 5).****To improve the quality and resolution of Figure 1 and consider its organization and presentation.****Address the kinetical and structural concerns pointed, including  the generation of 3D models of WT and mutants of PhZ employing AlphaFold.**Please submit your revised manuscript by Oct 07 2024 11:59PM, If you will need more time than this to complete your revisions, please reply to this message or contact the journal office at plosone@plos.org. Please include the following items when submitting your revised manuscript:A rebuttal letter that responds to each point raised by the academic editor and reviewer(s). You should upload this letter as a separate file labeled 'Response to Reviewers'.A marked-up copy of your manuscript that highlights changes made to the original version. You should upload this as a separate file labeled 'Revised Manuscript with Track Changes'.An unmarked version of your revised paper without tracked changes. You should upload this as a separate file labeled 'Manuscript'.

We look forward to receiving your revised manuscript.

Kind regards,

Mario Pedraza-Reyes, Ph.D.

Academic Editor

PLOS ONE

**Journal Requirements:**

Agencia Nacional de Investigación e

Innovación under the codes

PR_FCE_3_2013_1_100659 (MBV) and POS_1097576 (MBV).

Comisión Académica de Posgrado CAP_2019 (MBV)

3. Please upload a new copy of Figure 1 as the detail is not clear. Please follow the link for more information: " ext-link-type="uri" xlink:type="simple">https://blogs.plos.org/plos/2019/06/looking-good-tips-for-creating-your-plos-figures-graphics/"
" ext-link-type="uri" xlink:type="simple">https://blogs.plos.org/plos/2019/06/looking-good-tips-for-creating-your-plos-figures-graphics/"

Reviewers' comments:

Reviewer's Responses to Questions

**Comments to the Author**

1. Is the manuscript technically sound, and do the data support the conclusions?

Reviewer #1: Partly

Reviewer #2: Yes

Reviewer #3: Yes

2. Has the statistical analysis been performed appropriately and rigorously? 

Reviewer #1: Yes

Reviewer #2: N/A

Reviewer #3: No

3. Have the authors made all data underlying the findings in their manuscript fully available?

Reviewer #1: Yes

Reviewer #2: Yes

Reviewer #3: Yes

4. Is the manuscript presented in an intelligible fashion and written in standard English?

Reviewer #1: Yes

Reviewer #2: Yes

Reviewer #3: Yes

5. Review Comments to the Author

**Reviewer #1:** Comments to article PONE-D-24-30056

The article PONE-D-24-30056 evaluates, by mutational analysis, the amino acid residues important for purine transport in the PhZ protein of Phanerochaete chrysosporium. This work is based on the premise that the NAT transporter family has 2 different types of protein groups, the UapA/C-like ones that are specific for the transport of uric acid, xanthine, oxypurinol and allopurines and the AzgA-like ones that are transporters of hypoxanthine, guanine, 8-azaguanine, 6-mercaptopurine. Based on structural comparisons of AzgA-like proteins, conserved motifs in proteins of different taxonomic ranks are chosen to make changes in residues that could participate in purine transport, choosing the phZ gene to make the site-directed mutations that lead to the desired amino acid changes. Both the unmutated PhZ protein (WT) and mutant versions of the PhZ protein were heterologously expressed in an Aspergillus nidulans strain with mutations in purine transporters (ΔZAC). In this genetic background, the ability of these mutant proteins to transport purines was evaluated by assessing the growth of strains with purines as the sole nitrogen source, the uptake of radiolabeled hypoxanthine, the intracellular localization of the fusions of the WT PhZ protein and its variants, and western blot analysis. As a result of all these analyses, three-dimensional models of the PhZ protein with hypoxanthine and 8-azaguanine are established.

The approach of this work is very interesting; however, it is important to address certain inconsistencies in some points of the manuscript in order to transmit the information more efficiently.

General remarks

1. The aim of the study should be made clear, as well as the reason why PhZ was chosen for mutational analysis, as this is very ambiguous in the introduction and leads to many questions that could affect the interpretation of the overall work.

2. The description of the results should be carefully revised and rewritten. At several points in the manuscript, the description of the results seems more like a methodological description and in the figure captions the results are described.

3. In the results of the structural model, the description should focus more on the results obtained and not on comparisons with other models, as the notion of the real contribution of the work is lost. Comparisons with the results of other authors should be moved to the discussion.

4. Express decimal amounts with periods and not commas, for example, 0.8 instead of 0,8 for clarity.

Specific comments

Line 34. Replace "purines" with "purine".

Line 74. Has the structure of the AzgA protein of Aspergillus nidulans or any other microorganism been previously crystallized?

Lines 91-104. The objective of the study is not well defined, so the following questions are raised:

1. Why was the mutational analysis not performed directly on the AzgA protein of Aspergillus nidulans? Is its structure and the residues involved in purine transport already known? If so, this should be indicated in the introduction.

2. Why was PhZ chosen among the 11 AzgA-type proteins? What is the importance of analyzing this protein?

If the goal is to understand the structural differences between UapA/C and AzgA proteins that lead to substrate differences, as inferred in lines 82-84, why not perform a mutational analysis of UapA/C residues involved in purine transport, or of differential residues with AzgA, if the crystal structure of UapA/C is known

Lines 91-114. I recommend rewriting this paragraph to give fewer details of the results and to concisely describe the contribution of the work.

Lines 111-112. The hypothesis is very ambiguous

Table S1. Correct the description of the strain phZ_V58A*.

Line 126. Put the formula for ammonium tartrate in parentheses, not just ammonium.

Line 127. Indicate which purines and where they were acquired from.

Line 128. Indicate why these concentrations of 8-azaguanine and oxypurinol were used to assess toxicity. If a reference exists as background, include it. If not, provide adequate justification.

Line 166. Recommended by...?

Line 257. The representation of the conserved motifs in a table in Figure S2 is very confusing and difficult to understand. I suggest generating a figure with the alignment of the sequences.

Line 264. The figure caption in Figure 1 is too long. It should be more concise and describe the figure in the text in more detail instead of too much description in the figure caption. I suggest splitting Figure 1 into 2 parts so that Figure 1 B is Figure 2 so that it can be analyzed separately as it discusses the mutants generated.

Line 303-311. Part of the information in this section corresponds to the methodology.

Line 334. I consider that figure 2 should be better described in the text, not only in the figure caption, since the figure caption indicates why only the photos of the hx results are included and not those of adenine and guanine, but this is not mentioned in the text, or I suggest changing figure 2 for figure S3 but including the control in which AzgA is expressed.

Photographs of the growth of the WT strain of Aspergillus nidulans under the different conditions should also be included in this figure.

Line 372. Fig.3. Indicate what %V means.

Line 400. Indicate with which substrate differences were observed.

**Reviewer #2: **The authors present a complete characterization of several single mutants of purine transporter from Phanerochaete chrysporium. The manuscript will be of interest to those involved in the study of purine transporters (substrate specificity and transport efficiency) and in general in protein structure and function analysis. The work was performed applying cutting-edge techniques that supports the findings and conclusions stated. However, there are minor details that deserve attention.

1.- The IC50 value depends on substrate concentration used, therefore, how the concentration of radiolabeled substrate used (at least 10 times lower than the Km value) affected the Ki determination?

2.- In the same context, please explain in the text the meaning of Km/i and include a table as supplementary material with the values of IC50 obtained and indicate the value of Km used.

3.- Why do not try to use an AI software such as Alphafold to generate the 3D model of the transporter?

4.- Please include the stereochemical evaluation of the final model used for structural and docking studies.

5.- If the protein is a transporter, why did not perform the MD simulation using a membrane instead of an aqueous system?

6.- In would be desirable to generate the corresponding 3D models at least from the most interesting mutants to perform docking studies to explain the experimental data and not only with the native protein structure.

**Reviewer #3: **In this work, the authors describe a structure-function study of the purine transporter PhZ from Phanerochaete chrysosporium, initially focused on 4 conserved motifs of the NAT family, that unveiled a set of amino acid residues critical for transport activity and substrate specificity.

General comments:

This work utilizes a PhZ 3D model constructed by homology modeling using the crystal structure of UapA, further refined by molecular dynamics. Given the recent advances in 3D structure prediction by AlphaFold it would be interesting to include some data on this model, and compare it with the model used in this work. If the authors have already tested it, they could include some information in the manuscript. If not, they could retrieve the predicted structure by AlphaFold2 at https://colab.research.google.com/github/sokrypton/ColabFold/blob/main/AlphaFold2.ipynb.

It is important that the authors include statistical analysis of the data to determine if, in fact, there are significant differences in some of the presented results, as sometimes stated in the manuscript.

Abstract

The authors should state here why the A418 residue is pivotal and its location in the protein.

Introduction

Line 48: Please clarify if AzgA subfamily lacks homologs in animals or structural homologs

Line 84: Papakostas and colleagues (2013), the authors should include the reference number

Line 87-90: please include here what were the main findings of this study, as they are important for the work presented here.

Line 108: The reference regarding the post-translational regulation of the PhZ transporter is missing.

Materials and Methods

Line 177: Why weren’t the Km values calculated directly with the labeled substrate at different concentrations? Why did the authors prefer to use a fixed concentration of labelled hypoxanthine and various concentrations of unlabelled hypoxanthine to determine the Ki?

Line 186: What was the specific activity of the labelled hypoxanthine?

Results

Line 246: The authors should include the accession of all the sequences used in the alignment in the supplementary data.

Lines 259-262: where are located motifs 1 and 3? This will help the reader understand why this study discarded these motifs.

Figure 1 B) This image could be improved as several mutations overlap. Instead of showing the complete alignment, repeating the previous image, the authors could use a simplified scheme where the position of each amino acid residue is reported regarding the PhZ protein, evidencing the effect of each mutation.

Line 305: Please clarify here the name of the strain and the main purine transporters deleted.

Line 305: The authors state that the GFP did not affect the transporter activity, as the same growth phenotypes were observed in previous work. However, did the authors test the kinetic parameters of hypoxantine transport without the C- terminus GFP fusion? This method is more accurate to determine if the GFP tag affects protein activity.

Line 319: Instead of “ less than that of the wild-type” consider “ is less than that of cells expressing the wild-type”

line 330: include the growth phenotypes of this strain as mentioned for the previous strains.

line 360: modify the efficiency is not very clear, as it can be an increase or decrease in transport activity. In this case the phenotype suggests an increase in transport capacity.

line 361: please state the hypoxanthine concentration used to determine the initial uptake rate, and why this concentration was chosen.

line 368: Statistical analysis should be applied to this data to evaluate which are significantly different.

line 385 (figure 4): no SD are visible in the data concerning the uptakes of the L214M mutant. Is this correct?

line 389: The authors state that this mutant has lower transport capacity, but they only report the assay where they tested one concentration. The Vmax of the transporter was not determined to evaluate transporter capacity.

lines 394-397: This depends on the concentration of the unlabelled substrate. What were the conditions tested?

Line 415 (Fig 5) I consider it not easy to visualize the degree of inhibition of each substrate for the different mutants. I believe it would be easier if the data were presented differently, first for the WT and then for each mutant as follows: first, the V of hypoxanthine uptake w/o any inhibitor (100%), then with unlabelled hypoxanthine, and then the corresponding V % for all the tested inhibitors.

Also, statistical analysis should be presented here.

Discussion:

Line 616: Please explain how an increase in the intrinsic stability of the mutant protein would decrease the hypoxanthine uptake rate.

6. PLOS authors have the option to publish the peer review history of their article (what does this mean?). If published, this will include your full peer review and any attached files.

Reviewer #1: No

Reviewer #2: No

Reviewer #3: No

---

## [Author Response · Author response to Decision Letter 0]

14 Oct 2024

Dear Editor,

Regarding manuscript ID PONE-D-24-30056 - Mutational analysis of Phanerochaete chrysosporium´s purine transporter, the authors would like to thank you for considering the publication of our work and present the revised version in which all comments have been taken into account. 

Academic editor.

• Reorganize the introduction section and concisely describe the main objective and contribution of the work.

• Avoid repeating the experimental approaches in the Results section and rather focus in describing the results obtained.

• Some caption figures need to be rewritten to summarize the information (Figure 1) and/or clarifying specific elements (Figs. 3 and 5).

• To improve the quality and resolution of Figure 1 and consider its organization and presentation.

As suggested, the Introduction, Results and Materials Methods sections were reorganised. The figures were also improved for the new version. Figure 1 (A and B) was separated into two independent figures (Figure 1 and Figure 2). Both new figures, as well as Figures 3 (Figure 4 in the new version) and 5 (Figure 6 in the new version), were redone and reorganised, improving quality and clarity. Each caption was revised and abbreviated.

• Address the kinetical and structural concerns pointed, including the generation of 3D models of WT and mutants of PhZ employing AlphaFold.

All of the kinetical concerns raised by the reviewer have been addressed. The statistical analysis has been performed as requested. The final results are conceptually consistent, with minor variations in the values resulting from differences between analysis methodologies. In the previous analysis, samples from each of the two trials performed were normalised to their respective wild-type values (3 replicates each) and then the results from both experiments were averaged. In the current analysis, all trials are analysed together (6 replicates). As a contribution of the new analysis, it was confirmed that the small differences in transport between some of the mutants and the wild-type PhZ strain were statistically significant for 8-azaguanine (as previously reported) but also for all other physiological substrates.

When we constructed the initial model in 2020, AlphaFold was still in its earlier stages and had not yet achieved the high accuracy it's known for today. As requested, we generated a new model using the latest version of AlphaFold and compared it with our initial homology model. The two models were similar, with an RMSD of 3.35 Å, but the AlphaFold model showed better stereochemical quality, particularly in the loop regions. For this reason, we decided not to present the results from the homology model and have used only the AlphaFold model for the revised docking calculations.

We appreciate the reviewers' comments, which, as outlined above, have been fully addressed. All sections, particularly the Introduction and Results (including figure captions), have been thoroughly corrected and reorganized in accordance with the reviewers' suggestions. We believe that these changes have significantly enhanced the quality of the manuscript and the presentation of the results. We are grateful for the constructive feedback and hope that the revisions meet the journal's standards and adequately respond to the reviewers' suggestions.

Sincerely,

Mariana Barraco-Vega and Manuel Sanguinetti

Journal Requirements:

1. When submitting your revision, we need you to address these additional requirements. Please ensure that your manuscript meets PLOS ONE's style requirements, including those for file naming. The PLOS ONE style templates can be found at 

Done.

Agencia Nacional de Investigación e Innovación under the codes PR_FCE_3_2013_1_100659 (research grant MBV) and POS_1097576 (doctoral fellowship MBV).

Comisión Académica de Posgrado CAP_2019 (doctoral finalisation fellowship MBV)

Please state what role the funders took in the study.

3. Please upload a new copy of Figure 1. Done.

4. PLOS ONE now requires that authors provide the original uncropped and unadjusted images underlying all blot or gel results reported in a submission’s figures or Supporting Information files. 

The original image of the Western blot is provided as Supplementary Information (S1_raw_image).

Reviewer #1: Comments to article PONE-D-24-30056

General remarks

1. The aim of the study should be made clear, as well as the reason why PhZ was chosen for mutational analysis, as this is very ambiguous in the introduction and leads to many questions that could affect the interpretation of the overall work.

We have revised the introduction to clarify the aim of the study and explicitly stated the reasons for choosing PhZ for mutational analysis. This modification addresses the ambiguity and provides a clearer rationale for its selection.

2. The description of the results should be carefully revised and rewritten. At several points in the manuscript, the description of the results seems more like a methodological description and in the figure captions the results are described.

We have carefully revised and rewritten the description of the results to ensure they are clearly presented. We have separated methodological details from the results, and the figure captions now focus solely on describing the figures.

3. In the results of the structural model, the description should focus more on the results obtained and not on comparisons with other models, as the notion of the real contribution of the work is lost. Comparisons with the results of other authors should be moved to the discussion.

We have revised the results section to ensure that the contribution of our work is clearly highlighted. While the comparison of our results with those from other studies is discussed in more depth in the discussion section, we believe it is necessary to include some mentions in the results section to validate the model obtained.

4. Express decimal amounts with periods and not commas, for example, 0.8 instead of 0,8 for clarity. 

Done.

Specific comments

Line 34. Replace "purines" with "purine. Done.

Line 74. Has the structure of the AzgA protein of Aspergillus nidulans or any other microorganism been previously crystallized? 

A sentence was added in this regard.

Lines 91-104. The objective of the study is not well defined, so the following questions are raised:

1. Why was the mutational analysis not performed directly on the AzgA protein of Aspergillus nidulans? Is its structure and the residues involved in purine transport already known? If so, this should be indicated in the introduction.

2. Why was PhZ chosen among the 11 AzgA-type proteins? What is the importance of analyzing this protein? If the goal is to understand the structural differences between UapA/C and AzgA proteins that lead to substrate differences, as inferred in lines 82-84, why not perform a mutational analysis of UapA/C residues involved in purine transport, or of differential residues with AzgA, if the crystal structure of UapA/C is known

Answers 1 and 2. These were addressed in the new version of the manuscript. Responding to the reviewer, we would like to say that although part of our group was responsible for the identification of the A. nidulans AzgA protein and its first characterisation, our research is currently focused on basidiomycetes fungi. In this study, then, we chose to work with PhZ to extend studies of the AzgA-like subfamily to Basidiomycetes since most studies focus on Ascomycetes. Additionally, the requested information regarding the structure of AzgA and the residues involved in purine transport has been incorporated into the introduction of the revised manuscript. While one of the objectives is to understand the structural differences between UapA/C and AzgA proteins that lead to differences in substrate specificity, in this work, we did not focus on analysing the UapA/C residues involved in purine transport, as this approach was already addressed in Krypotou et al. 2014. The aim of this study is to contribute to the characterization of the AzgA-like family from a different perspective. Further clarification on the importance of analysing PhZ has been incorporated into the new version of the manuscript.

Lines 91-114. I recommend rewriting this paragraph to give fewer details of the results and to concisely describe the contribution of the work. Done.

Lines 111-112. The hypothesis is very ambiguous. 

We hope to have better defined the hypothesis.

Table S1. Correct the description of the strain phZ_V58A*. Done.

Line 126. Put the formula for ammonium tartrate in parentheses, not just ammonium. Done.

Line 127. Indicate which purines and where they were acquired from. Done.

Line 128. Indicate why these concentrations of 8-azaguanine and oxypurinol were used to assess toxicity. If a reference exists as background, include it. If not, provide adequate justification. 

Done. We added the reference.

Line 166. Recommended by...? Added.

Line 257. The representation of the conserved motifs in a table in Figure S2 is very confusing and difficult to understand. I suggest generating a figure with the alignment of the sequences. 

Figure S2 has been reorganised for easier reading and understanding. A file was generated with the identification of the sequences used (S1 Appendix). 

Line 264. The figure caption in Figure 1 is too long. It should be more concise and describe the figure in the text in more detail instead of too much description in the figure caption. I suggest splitting Figure 1 into 2 parts so that Figure 1 B is Figure 2 so that it can be analyzed separately as it discusses the mutants generated. Done. 

Line 303-311. Part of the information in this section corresponds to the methodology. Done. 

Line 334. I consider that figure 2 should be better described in the text, not only in the figure caption, since the figure caption indicates why only the photos of the hx results are included and not those of adenine and guanine, but this is not mentioned in the text, or I suggest changing figure 2 for figure S3 but including the control in which AzgA is expressed. Photographs of the growth of the WT strain of Aspergillus nidulans under the different conditions should also be included in this figure.

A sentence describing both, what is shown in each Figure 2 (Fig 3 in the new version) and S3 Fig, and the criteria used, was added to the text. The description of the results observed in both figures was also completed in the text. 

Line 372. Fig.3. Indicate what %V means. Done. 

Line 400. Indicate with which substrate differences were observed. 

This part has been reorganised to clarify the description. 

Reviewer #2: 

The authors present a complete characterization of several single mutants of purine transporter from Phanerochaete chrysporium. The manuscript will be of interest to those involved in the study of purine transporters (substrate specificity and transport efficiency) and in general in protein structure and function analysis. The work was performed applying cutting-edge techniques that supports the findings and conclusions stated. However, there are minor details that deserve attention.

1.- The IC50 value depends on substrate concentration used, therefore, how the concentration of radiolabeled substrate used (at least 10 times lower than the Km value) affected the Ki determination?

As recommended in Krypotou et al. 2014, the Km/i is calculated using the formula Ki = IC50 / (1 + [S]/Km), where [S] is the concentration of the radiolabelled substrate. In our assays, Ki values equal IC50 values since the [S] is very low (at least 10-fold lower than the Km value). The method is in fact identical to the one for Km determination, but stock solutions are prepared using a mixture of fixed radiolabelled substrate and increasing concentrations of non-radiolabelled substrates.

2.- In the same context, please explain in the text the meaning of Km/i and include a table as supplementary material with the values of IC50 obtained and indicate the value of Km used.

In the revised manuscript, we have included an explanation of the meaning of Km/i within the text. The values of Km/i, which under our experimental conditions are equivalent to IC50, are presented in Figure 5. Therefore, we have not included an additional supplementary table, as these data are already displayed in the figure.

3.- Why do not try to use an AI software such as Alphafold to generate the 3D model of the transporter? 

4.- Please include the stereochemical evaluation of the final model used for structural and docking studies. 

Answer 3 and 4. When we constructed the initial model in 2020, AlphaFold was still in its earlier stages and had not yet achieved the high accuracy it's known for today. Back then, homology modelling was a widely accepted and reliable approach for generating 3D models, especially for complex systems like transporters. As requested, we generated a new model using the latest version of the software and compared it with our initial model. Upon comparison, we found that the RMSD (Root Mean Square Deviation) between the AlphaFold model and the original is 3.35 Å, with the largest differences mainly located in the loop regions, suggesting that our first model was quite close to the actual structure.

Finally, as requested, we performed a stereochemical comparison of both models using PROCHECK. Most residues fall within favourable regions of the Ramachandran plot, and the G-factors are close to zero, indicating good overall stereochemical quality. The AlphaFold model has slightly better values than our initial one (see S5 Fig in the manuscript and table below).

Taking it all together, we decided it is unnecessary to include both models in the manuscript, and therefore, we will only present the AlphaFold model, redoing the docking calculations accordingly.

5.- If the protein is a transporter, why did not perform the MD simulation using a membrane instead of an aqueous system?

The molecular dynamics (MD) simulation was performed with the primary goal of refining the model and improving its structural quality, rather than studying the protein’s mechanism or transport function. For model refinement purposes, a simpler aqueous system was sufficient. Using a membrane environment would be more appropriate if the objective were to investigate the protein's interactions with the lipid bilayer, its transport mechanism, or other functional aspects. In this case, the focus was solely on optimizing the structural features of the model

6.- In would be desirable to generate the corresponding 3D models at least from the most interesting mutants to perform docking studies to explain the experimental data and not only with the native protein structure.

Done. We generated AlphaFold model and performed docking studies for the L124 mutant, as this residue is part of the binding site and exhibited significant differences in transport activity. In contrast, I388, which was also considered, did not show any differences in transport and was therefore not included in the docking analysis. 

Reviewer #3: 

In this work, the authors describe a structure-function study of the purine transporter PhZ from Phanerochaete chrysosporium, initially focused on 4 conserved motifs of the NAT family, that unveiled a set of amino acid residues critical for transport activity and substrate specificity.

General comments:

This work utilizes a PhZ 3D model constructed by homology modeling using the crystal structure of UapA, further refined by molecular dynamics. Given the recent advances in 3D structure prediction by AlphaFold it would be interesting to include some data on this model, and compare it with the model used in this work. If the authors have already tested it, they could include some information in the manuscript. If not, they could retrieve the predicted structure by AlphaFold2.

As requested, we generate

---

## [Editor Report · Decision Letter 1]

21 Oct 2024

Mutational analysis of Phanerochaete chrysosporium´s purine transporter

PONE-D-24-30056R1

Dear Dr. M. Sanguinetti

We’re pleased to inform you that your manuscript has been judged scientifically suitable for publication and will be formally accepted for publication once it meets all outstanding technical requirements.

Kind regards,

Mario Pedraza-Reyes, Ph.D.

Academic Editor

PLOS ONE

---

## [Editor Report · Acceptance letter]

23 Oct 2024

PONE-D-24-30056R1 

PLOS ONE

Dear Dr. Sanguinetti, 

I'm pleased to inform you that your manuscript has been deemed suitable for publication in PLOS ONE. Congratulations! Your manuscript is now being handed over to our production team.

Kind regards, 

on behalf of

Dr. Mario Pedraza-Reyes 

Academic Editor

PLOS ONE